# Dynamic changes of Bacterial Microbiomes in Oropharynx during Infection and Recovery of COVID-19 Omicron Variant

Guangying Cui[1☯], Ying Sun[2☯], Yawen Zou[3☯], Ranran Sun[1☯], Yanxia Gao[4], Xiaorui Liu[1], Yongjian Zhou[1], Donghua Zhang[5], Xueqing Wang[6], Yonghong Li[5], Liwen Liu[1], Guizhen Zhang[1], Benchen Rao[1], Zujiang Yu[1]*, Zhigang Ren[1]*

1 Department of Infectious Diseases, State Key Laboratory of Antiviral Drugs, Pingyuan Laboratory, the First Affiliated Hospital of Zhengzhou University, Zhengzhou, China, 2 State Key Laboratory for Diagnosis and Treatment of Infectious Diseases, National Clinical Research Center for Infectious Disease, National Medical Center for Infectious Diseases, Collaborative Innovation Center for Diagnosis and Treatment of Infectious Diseases, The First Affiliated Hospital, College of Medicine, Zhejiang University, Hangzhou, China, 3 Bio-X Institutes, Key Laboratory for the Genetics of Developmental and Neuropsychiatric Disorders (Ministry of Education), Shanghai Jiao Tong University, Shanghai, China, 4 Emergency Department, the First Affiliated Hospital of Zhengzhou University, Zhengzhou, China, 5 Anyang City Fifth People's Hospital, Long An District, Anyang, China, 6 Reproductive Medicine Center, the First Affiliated Hospital of Zhengzhou University, Zhengzhou, China

☯ These authors contributed equally to this work.
* fccrenzg@zzu.edu.cn (ZY); johnyuem@zzu.edu.cn (ZR)

**Data Availability Statement:** We have adjusted the release time of the data. The metagenomics sequencing dataset of the above study was deposited in The National Center for Biotechnology

## Abstract

Oropharyngeal microbiomes play a significant role in the susceptibility and severity of COVID-19, yet the role of these microbiomes play for the development of COVID-19 Omicron variant have not been reported. A total of 791 pharyngeal swab samples were prospectively included in this study, including 297 confirmed cases of Omicron variant (CCO), 222 confirmed case of Omicron who recovered (CCOR), 73 confirmed cases of original strain (CCOS) and 199 healthy controls (HC). All samples completed MiSeq sequencing. The results showed that compared with HC, conditional pathogens increased in CCO, while acid-producing bacteria decreased. Based on six optimal oropharyngeal operational taxonomy units (OTUs), we constructed a marker microbial classifier to distinguish between patients with Omicron variant and healthy people, and achieved high diagnostic efficiency in both the discovery queue and the verification queue. At same time, we introduced a group of cross-age infection verification cohort and Omicron variant subtype XBB.1.5 branch, which can be accurately distinguished by this diagnostic model. We also analyzed the characteristics of oropharyngeal microbiomes in two subgroups of Omicron disease group—severity of infection and vaccination times, and found that the change of oropharyngeal microbiomes may affect the severity of the disease and the efficacy of the vaccine. In addition, we found that some genera with significant differences gradually increased or decreased with the recovery of Omicron variant infection. The results of Spearman analysis showed that 27 oropharyngeal OTUs were closely related to 6 clinical indexes in CCO and HC. Finally, we found that the Omicron variant had different characterization of oropharyngeal microbiomes from the original strain. Our research characterizes oropharyngeal microbiomes of Omicron

Information at https://www.ncbi.nlm.nih.gov/bioproject/PRJNA911036, accession number PRJNA911036. The metagenomics sequencing dataset about confirmed cases of original strain was deposited in The National Center for Biotechnology Information at https://www.ncbi.nlm.nih.gov/sars-cov-2/, accession number PRJNA739539.

**Funding:** This work was supported by the National Key Research and Development Program of China (2022YFC2303100 to ZR and 2023YFC3043514 to ZY); the Henan Province Epidemic Prevention and Control Emergency Scientific Research Project (221111311700 to ZR and 221111311600 to ZY); the Scientific Research and Innovation Team of The First Affiliated Hospital of Zhengzhou University (QNCXTD2023002 to ZR and ZYCXTD2023002 to ZY); and the Independent and Innovation Project for Graduate Students of Zhengzhou University (20230444 to BR). The funders had no role in study design, data collection and analysis, decision to publish, or preparation of the manuscript.

**Competing interests:** The authors have declared that no competing interests exist.

variant cases and rehabilitation cases, successfully constructed and verified the non-invasive diagnostic model of Omicron variant, described the correlation between microbial OTUs and clinical indexes. It was found that the infection of Omicron variant and the infection of original strain have different characteristics of oropharyngeal microbiomes.

## Author summary

COVID-19, caused by the SARS-CoV-2 virus, was first discovered in Wuhan, Hubei Province, China in December 2019, causing a huge impact around the world. In the past three years, novel coronavirus has successively appeared many varieties through gene mutation, among which Omicron strain is the most popular variety. Studies have shown that the imbalance of microflora is closely related to the progress of novel coronavirus infection and the severity of clinical symptoms. In the previous study, we build database of oral-intestinal microbial metagenomics, immunome and metabolome of novel coronavirus Original Strain with mild, medium and severe clinical symptoms and healthy subjects, reveal the role and mechanism of the oral and intestinal microbiome in Original Strain infection and exacerbation, and build the model of susceptibility, severe warning and prognosis of Original Strain pneumonia which based on the microbiome-immunity-metabolism. Our research characterizes oropharyngeal microbiomes of Omicron variant cases and rehabilitation cases, successfully constructed and verified the non-invasive diagnostic model of Omicron variant. We also analyzed the characteristics of oropharyngeal microbiomes in two subgroups of Omicron disease group—severity of infection and vaccination times, and found that the change of oropharyngeal microbiomes may affect the severity of the disease and the efficacy of the vaccine. And we described the correlation between microbial OTUs and clinical indexes. At last, it was found that the infection of Omicron variant and the infection of original strain have different characteristics of oropharyngeal microbiomes.

## 1. Introduction

Novel coronavirus disease (COVID-19), caused by severe acute respiratory syndrome coronavirus type 2, was first discovered in Wuhan, Hubei Province, China in December 2019. Due to the unprecedented speed of transmission and the extreme impact on public health, it has quickly become a serious problem of global concern. In past three years, due to the continuous changes of coronavirus through genetic mutation, there have been Alpha, Beta, Gamma, Delta, and the main circulation variety Omicron, which makes us face many challenges on the road of eliminating the virus. Of all the known variants of SARS-CoV-2, Omicron has the largest number of mutation sites, which may be related to heredity, disease severity and immune escape [1]. Studies have shown that the virulence of Omicron is weaker than that of other mutants, and the clinical symptoms after infection are also milder [2–5]. But it has a very strong transmission power, which may be related to a large number of abrupt changes in receptor binding domain (RBD) of Omicron and surface electrostatic potential [6–8]. On the other hand, the Omicron mutant may escape the immune protection of existing SARS-CoV-2 infection, which is one of the main reasons for its rapid spread [9]. This means that the current COVID-19 vaccine is unlikely to provide sufficient immunity to it [10], and people who have been infected with other SARS-CoV-2 variants in the past may be reinfected with this new

variant [11,12]. For this new variety, we need early and rapid detection of potential infections, especially asymptomatic infections, which is the focus of virus control.

The respiratory tract is the first barrier against pathogen invasion, but it is also the main route of transmission and replication of pathogens [13]. When the virus enters the host cell through the respiratory tract, it will not only cause inflammation and immune disorder, but also destroy the dynamic balance of the long-term colonized Microbiomes. We call the microbiomes of the respiratory tract "the gatekeeper of respiratory health", which plays an important role in the structural maturity of the respiratory tract and the formation of local immunity [14,15]. The imbalance of microbiomes are significantly related to the progression of SARS-CoV-2 and the severity of clinical symptoms [16]. The concept of using microbial markers as a non-invasive diagnostic tool for diseases has become more and more mature in recent years, including autoimmune liver disease [17], type 2 diabetes [18] and liver cirrhosis [19]. In our previous study, we successfully described the characteristics of oropharyngeal microbiomes of COVID-19 original strain and its recovered patients, and successfully constructed and verified the non-invasive diagnostic model for original strain [20]. However, with the emergence of Omicron Variant, the oropharyngeal microbial characteristics of patients infected with Omicron strain have not been reported. On January 8, 2022, two patients in Anyang City, Henan Province, China were diagnosed with SARS-CoV-2 infection. After sequencing the whole genome of the infected virus, it was confirmed that the Omicron variant BA.5.2 branch infection. Through epidemiological investigation and gene sequencing, it was found that the first outbreak in Henan was homologous to that in Tianjin. We thoroughly identified the infected population by using multiple nucleic acid monitoring and screening and tracking close contacts. A group of patients were diagnosed as Omicron variant infection by laboratory and sent to the COVID-19 designated hospital in Henan Province for centralized treatment. In this study, we described the characteristics of oropharyngeal microbiomes in patients with Omicron variant infection and recovered patients, and found out the differences in oropharyngeal microbiota between mild and moderate infections. By analyzing the imbalance of oropharyngeal microbiomes and the changes of marker bacteria, a non-invasive microbial diagnostic model was constructed and verified in varied populations. Finally, we clarified that both cases of Omicron variant and original strain had different characteristics of oropharyngeal microbiomes.

## 2. Results

### 2.1. Study Design and characteristics of the participants

We prospectively collected 835 oropharyngeal samples from Henan Province, China. After a rigorous screening and exclusion process, a total of 791 oropharyngeal samples were sequenced by 16S rRNA MiSeq (S1 Fig). Among them, 297 CCO samples and 199 HC samples were randomly divided into discovery queues (198 CCO and 133 HC) and verification queues (99 CCO and 66 HC). Then the characteristics of oropharyngeal microbiota were analyzed, the characteristics of microbiomes were described in the discovery queue, the microbiomes with obvious differences were found and the best bacteria were identified as diagnostic markers, and the diagnostic model of SARS-CoV-2 Omicron variety was constructed. In the verification phase, we use 99 CCO samples and 66 HC samples to verify the effectiveness of the diagnostic model, and it has been well verified in a group of infected cohorts of cross-age and a group of infected cohorts of different subtypes of Omicron variant.

The clinical characteristics of confirmed cases of Omicron and healthy controls in the discovery and validation cohort are shown in S2 Fig. The average age of confirmed cases of Omicron in the discovery phase was 29.92 years old, with a male-to-female ratio of 84: 114. The

average age of confirmed cases of Omicron in the verification phase was 30.25 years old, and the ratio of male to female was 42:57. At the same time, we also analyzed the blood routine and biochemical indexes of CCO and HC. Compared with HC, leukocytes (p <0.05), neutrophils (p<0.0001), platelets (p<0.01) and albumin (p<0.0001) decreased significantly in the confirmed cases of Omicron. The detailed clinical data of all subjects are shown in S1 and S2 Tables.

## 2.2. Oropharyngeal Microbial Diversity in SARS-CoV-2 Omicron Variant

In the discovery queue, the results of rarefaction analysis showed that the OTUs abundance of HC group and CCO group was nearly stable, and the OTUs richness of HC group was significantly higher than that of CCO group (Fig 1A). The results of Shannon index showed that the oropharyngeal microbial alpha diversity in HC group was significantly higher than that in CCO group (p<0.01) (Fig 1B and S3 Table). The results of PCoA and NMDS analysis showed that there was a significant difference in the distribution of oropharyngeal microbial community between CCO group and HC group, the overall microbial community composition

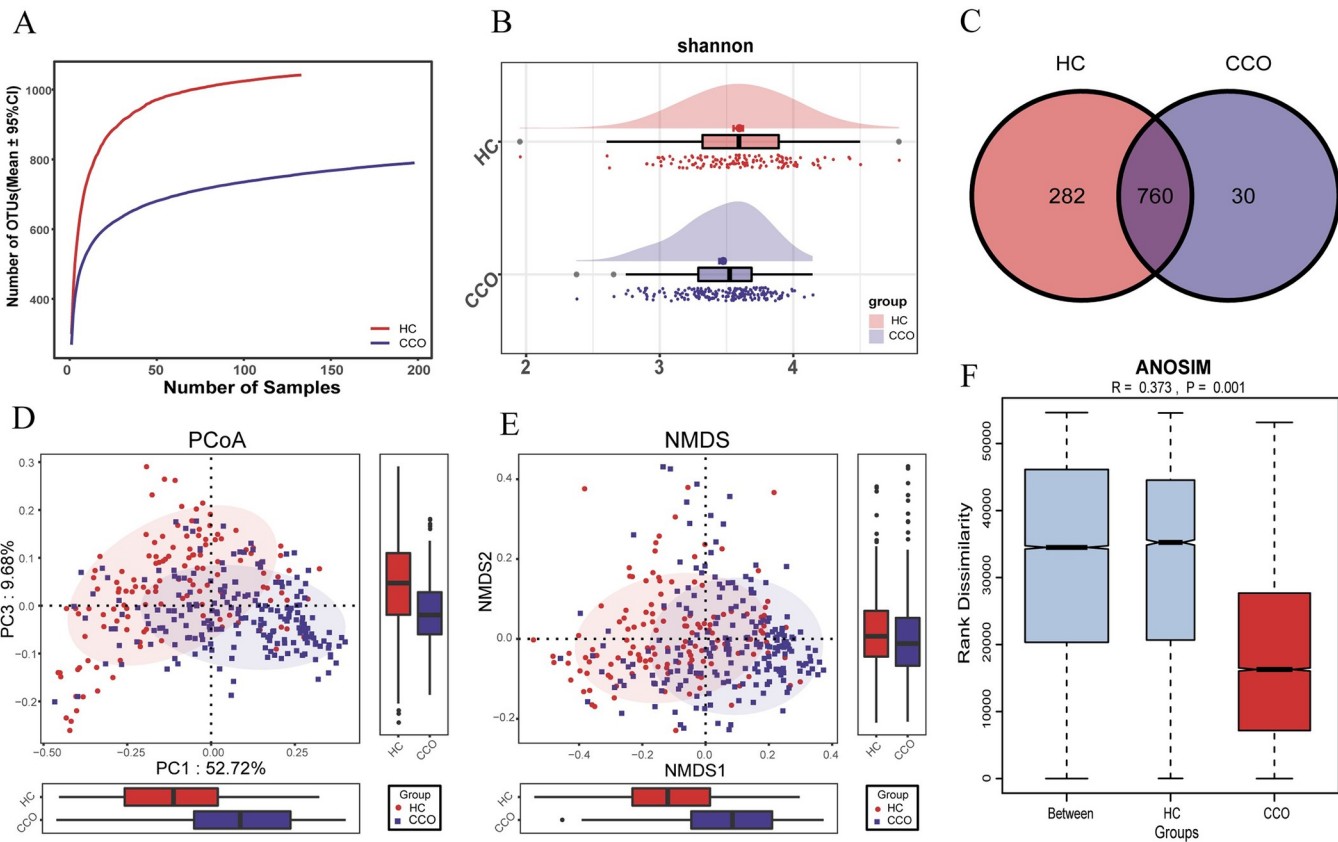

**Fig 1. The oropharyngeal microbial diversity between CCO and HC groups in the discovery cohort.** A) Rarefaction analysis showed as the number of samples raised, the number of OTUs approached saturation in CCO (n = 198) and HC (n = 133). The number of OTU in CCO group was significantly lower than that in HC group. B) Shannon index showed that the oropharyngeal microbial α-diversity in CCO group was significantly lower than that in HC group (p<0.01). C) A Venn diagram displaying the overlaps between groups showed that 760 of 1072 OTUs were shared in CCO and HC groups, while 30 of 1072 OTUs were unique to the CCO group. D) and E) The PCoA and NMDS based on OTU distribution showed the oropharyngeal taxonomic composition was significantly different between the two groups. F) The ANOSIM analysis in the form of boxplot shows that the difference between CCO and HC groups is greater than that within groups (R = 0.373, P = 0.001). CCO, confirmed cases of COVID-19 Omicron variant; HC, healthy controls; OTUs, operational taxonomy units; PCoA, principal coordinate analysis; NMDS, nonmetric multidimensional scaling; centerline, median; box limits, upper and lower quartiles; error bars, 95% CI.

changed between the two groups (Fig 1D–1F). In summary, we found that compared with the healthy population, there is an imbalance in the oropharyngeal microbiota of COVID-19 patients infected with Omicron strain. The Venn diagram shows that out of a total of 1072 OTUs, 760 OTUs are shared between the CCO group and the HC group, 282 OTUs are unique to the HC group, and 30 OTUs are unique to the CCO group (Fig 1C).

## 2.3. Phylogenetic profiles of oropharyngeal microbial communities in SARS-CoV-2 Omicron Variant

In the discovery stage, we further identified the microbial composition and changes of oropharyngeal microbiomes in CCO group and HC group. The results showed that *Bacteroidota*, *Firmicutes*, *Proteobacteria*, *Fusobacteriota* and *Actinobacteriota* was the five dominant bacteria in two groups, and its sum accounted for 97.5% of the total sequence in CCO group and 96.5% in HC group (Fig 2A and S4 Table). Next, we carried out Wilcoxon rank-sum test on the differential expression of two groups of bacteria at the phylum level. As a result, the 5 phyla were increased in CCO group compared with HC group (p<0.05), while the 11 phyla were decreased (p<0.05) (Fig 2B and S8 Table).

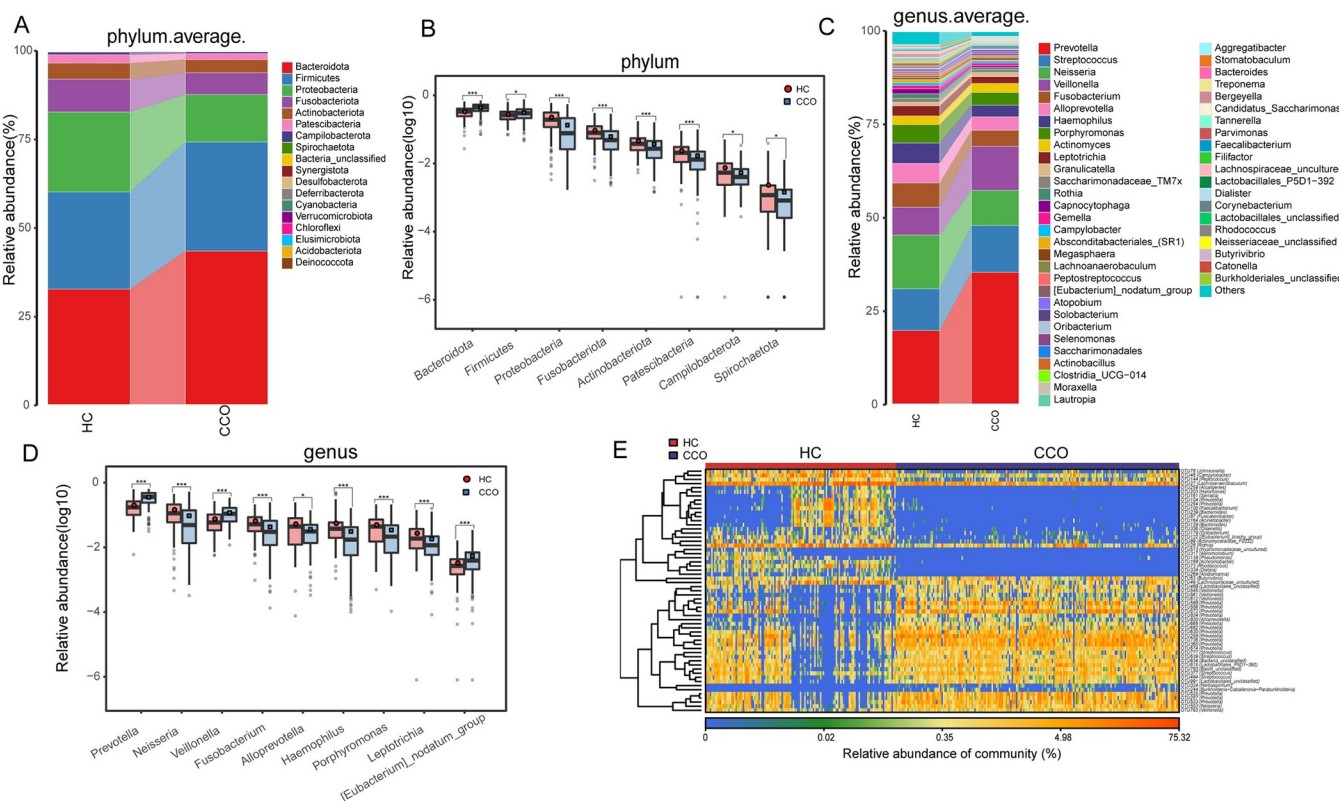

**Fig 2. Comparison of the composition and abundance of oropharyngeal microbiome between CCO (n = 198) and HC (n = 133) groups.** A) Average compositions and relative abundance of the bacterial community at the phylum level between CCO and HC groups. B) Compared with HC group, 5 phyla increased significantly and 11 phyla decreased significantly in CCO group (Only part of the bacteria is displayed. Please see the S8 Table for details). C) Average compositions and relative abundance of the bacterial community at the genus level between CCO and HC groups. D) Compared with HC group, 10 genera increased significantly and 53 genera decreased significantly in CCO group (Only part of the bacteria is displayed. Please see the S9 Table for details). E) Heatmap showed the relative abundances of differential OTUs for each sample between CCO and HC groups. *p<0.05, **p<0.01, ***p<0.001. CCO, confirmed cases of COVID-19 Omicron variant; HC, healthy controls; OTUs, operational taxonomy units; centerline, median; box limits, upper and lower quartiles; circle or square symbol, mean; error bars, 95% CI.

The average composition and relative abundance of oropharyngeal microbes at the genus level are shown in the Fig 2C. In the CCO group, the abundance of the main genera from high to low was in the order of *Prevotella*, *Neisseria*, *Streptococcus*, *Veillonella*, and *Fusobacterium*, and its sum accounted for 59.33% of the microbiota. In the HC group, the abundance of the main genera from high to low was in the order of *Prevotella*, *Streptococcus*, *Veillonella*, *Neisseria*, and *Fusobacterium*, and its sum accounted for 73.41% of the microbiota (S6 Table). A total of 63 genera with significant differences in abundance were identified by Wilcoxon rank-sum test (p<0.05) (Fig 2D and S9 Table). Among them, 10 genera such as *Prevotella*, *Veillonella* and *Eubacterium nodatum* were increased in CCO group compared with HC group, while 53 genera such as *Neisseria*, *Fusobacterium* and *Alloprevotella* were decreased. Finally, we showed 59 OTUs significantly increased or decreased in the CCO group compared with the HC group in the form of a heatmap (Fig 2E and S10 Table). In summary, our study revealed the unique composition of oropharyngeal microbiome of confirmed cases of Omicron variant, which was characterized by the increase of some conditional pathogenic bacteria (microbes that normally exist in the human body cause infection under specific circumstances, such as the decrease of immune function, changes or disorders of settlement sites and so on) such as *Prevotella*, *Veillonella*, while the decrease of acidogenic bacteria such as *Fusobacterium*, *Alloprevotella*, which balanced metabolic function.

In addition, we used the method of LEfSe analysis to select specific bacterial taxa related to Omicron strains. The phylogenetic cladogram of oropharyngeal microbiomes showed that there were significant differences in abundance from phyla to genus between CCO and HC groups (S3A Fig and S11 Table). According to the Kyoto Encyclopedia of Genes and Genomes (KEGG) database, we predicted the metabolic pathway and some important metabolic modules of the microbial community through the sequence of 16S rRNA marker genes. We show 89 enrichment pathways with the most significant differences between CCO and HC. In CCO group, 47 functions such as NOD like receptor signaling pathway and Folate biosynthesis and Vitamin B6 metabolism were significantly enhanced, while in HC group, 42 functions such as Bacterial chemotaxis and Bacterial secretion system and Butanoate metabolism were significantly enhanced (S3B Fig and S12 Table).

## 2.4. Diagnostic potential of the oropharyngeal microbial classifier for SARS-CoV-2 Omicron Variant

In the discovery phase (198 CCO and 133 HC), we constructed a five-fold cross-validation random forest model to demonstrate the diagnostic efficacy of oropharyngeal microbial classifier in patients infected with Omicron variant. The results showed that six OTUs were the best microbial marker sets, and they could accurately distinguish between CCO group and HC group (Fig 3A and 3B). Then, we use six OTUs markers to calculate the probability of disease (POD) index for the discovery phase and the verification phase. In the discovery stage, the POD value of the CCO group was significantly higher than that of the HC group, with an AUC value of 98.52% (95% CI 97.09% to 99.94%, p<0.0001) (Fig 3C and 3D and S13 Table). The above data show that oropharyngeal microbial markers can effectively diagnose the infection with Omicron variant.

In the verification phase, we calculated and analyzed the verification queue (99 CCO and 66 HC) to further verify the effectiveness of the classifier in the diagnosis of Omicron variant. The results showed that the POD index of the CCO group was also significantly higher than that of the HC group, and the AUC value between the two groups was 98.42% (95% CI 96.95% to 99.88%, p<0.0001) (Fig 3E and 3F and S14 Table). The above results verify the accurate diagnostic efficacy of oropharyngeal microbial markers in infection with Omicron variant.

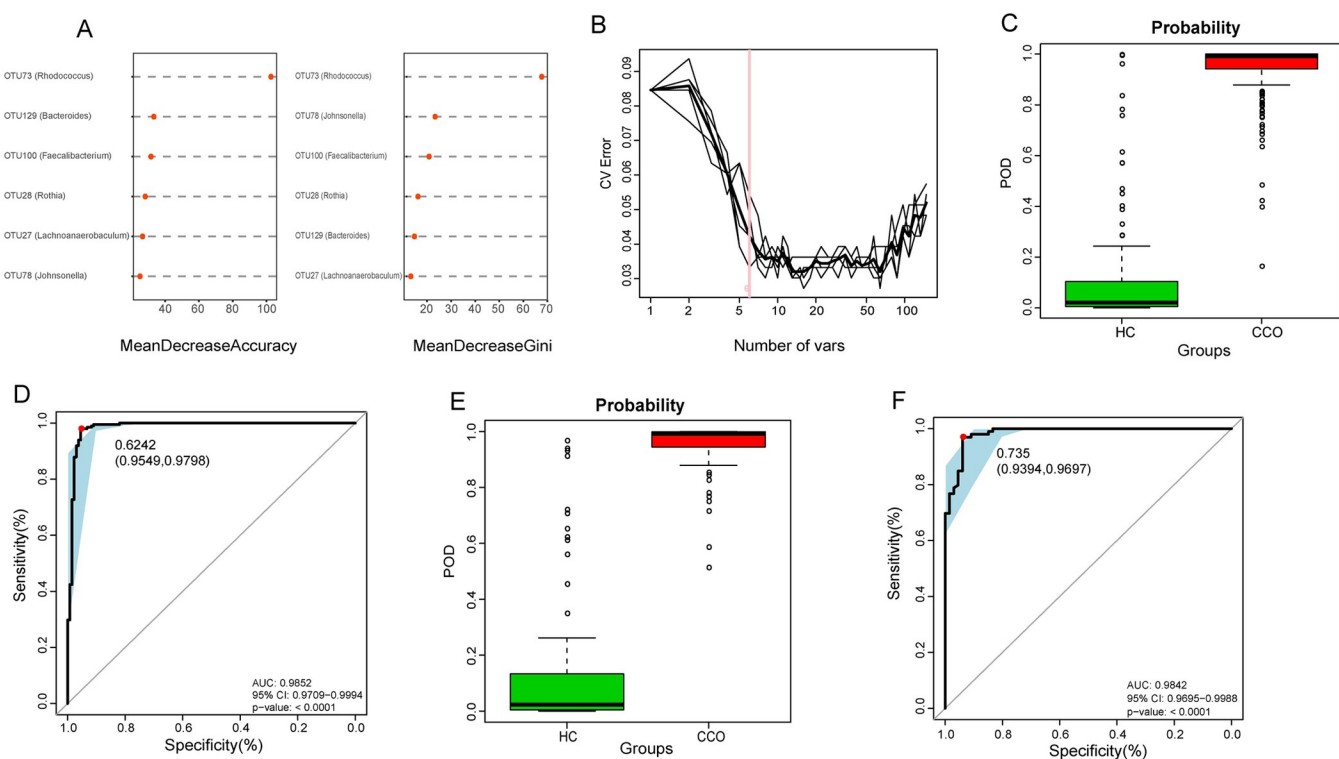

**Fig 3. Diagnostic efficacy of oropharyngeal microbial classifier on Omicron variant.** A) Six microbial markers were selected as the best markers set by random forest model. B) Importance distribution map of the selected microbial makers in the model. C) The POD value was significantly higher in CCO group (n = 198) compared with that in HC group (n = 133) in the discovery cohort. D) The POD value achieved an AUC of 98.52% (95% CI 97.09% to 99.94%, p<0.0001) between CCO group (n = 198) versus HC group (n = 133) in the discovery cohort. E) The POD value was significantly higher in CCO group (n = 99) compared with that in HC group (n = 66) in the verification cohort. F) The POD value achieved an AUC of 98.42% (95% CI 96.95% to 99.88%, p<0.0001) between CCO group (n = 99) versus HC group (n = 66) in the verification cohort. CCO, confirmed cases of COVID-19 Omicron variant; HC, healthy controls; OTUs, operational taxonomy units; POD, probability of disease; AUC, area under the curve; centerline, median; box limits, upper and lower quartiles; error bars, 95% CI.

In order to verify the universality of the diagnostic model, we included two cross-age cohorts for analysis. These included 20 confirmed cases of Omicron variant (CCO') with an average age of 53.15 years and 20 healthy controls (HC') with an average age of 57.65 years. The results showed that the POD index of the CCO' group was significantly higher than that of the HC' group, and the AUC value was 98.42% (95% CI 95.14% to 100%, p<0.0001) (S4A and S4B Fig). The above results verify that the diagnostic model is also suitable for the diagnosis of Omicron variant infection in the middle-aged and elderly.

In order to verify the specificity of the diagnostic model, we included oropharyngeal samples of 80 confirmed cases of Omicron subvariants XBB.1.5 (CCXBB), and distinguished the verification cohort in CCO group from CCXBB group by this diagnostic model. The results showed that the POD value of CCO group was significantly higher than that in CCXBB group, and the AUC value was 90.59% (95% CI 86.23% to 94.94%, p<0.0001) (S4C and S4D Fig). The above results prove that the diagnostic model can accurately distinguish between confirmed cases of Omicron subvariants BA.5.2 and confirmed cases of Omicron subvariants XBB.1.5.

## 2.5. Subgroup analysis of Omicron variant—Vaccination times and Severity of infection

The patients in the CCO group can be divided into mild and moderate types in terms of the severity of infection. Age, sex, clinical indicators (including white blood cells, neutrophils,

lymphocytes, hemoglobin, platelets, alanine aminotransferase, aspartate aminotransferase, albumin, total bilirubin, urea), vaccine injection times and other confounding factors were excluded by PSM (Propensity Score Matching), we selected 57 mild infection and 19 moderate infection oropharyngeal microbiomes for analysis. The results of Shannon index showed that there was no significant difference in microbial α-diversity between moderate and mild patients (S5A and S5B Fig, and S15 Table). The PCoA analysis showed that there were significant differences in the composition of oropharyngeal microbial community between the two groups (S5D Fig). Further analysis of the differences in the expression of oropharyngeal microbial between mild and moderate patients at different classification levels showed that there was no significant difference between the two groups of microbial at the phylum level (p>0.05). At the genus level, the *Streptobacillus*, *Lachnoanaerobaculum* and *Eubacterium Nodatum* of moderate patients was significantly higher than that of mild patients (p<0.05) (S5E Fig and S16 Table). In the Kyoto Encyclopedia of Genes and Genomes (KEGG) database, we found that there was no significant difference in metabolic pathways between mild and moderate patients.

By collecting the clinical data of 297 patients in the CCO group, we found that 277 patients received two times of COVID-19 vaccine and 20 patients received three times of COVID-19 vaccine. An interesting phenomenon is that the severity of the disease in patients who received three times of COVID-19 vaccine was mild, and 19 patients who received two times of COVID-19 vaccine had a moderate severity of the disease. In order to further study whether the efficacy of the vaccine is related to the imbalance of oropharyngeal microecology, we adopted the PSM by excluding confounding factors such as sex, age, severity of disease, white blood cells, neutrophils, lymphocytes, hemoglobin, platelets, alanine aminotransferase, aspartate aminotransferase, total bilirubin, urea and so on. 60 patients who received two times of COVID-19 vaccine (CCO2) and 20 patients who received three times of COVID-19 vaccine (CCO3) were selected to analyze the differential expression of bacterial species between the two groups by Willcoxon rank sum test. Our results show that there is no significant difference between the two groups at the phylum level (p>0.05). At the genus level, we find enrichment of *Granulicatella*, *lactobacillales* and *corynebacterium* in CCO2 compared with CCO3 (p<0.05) (S5F Fig and S17 Table).

## 2.6. Oropharyngeal microbial characterization among CCO, CCOR and HC

In order to find the potential microbiome involved in the infection recovery of Omicron variant, we further analyzed 297 CCO, 222 CCOR and 197 HC. We used rarefaction analysis of the samples in all groups to ensure the effectiveness of collecting samples. The OTUs richness of three groups were close to saturation. By using Shannon index analysis, the results showed that the α-diversity of CCO group and CCOR group was lower than that of HC group, and the CCOR group was lower than that of CCO group (p<0.05) (Fig 4A and S18 Table). The results of PCoA analysis showed that the microbial community distribution of CCOR group was significantly different from that of CCO group and HC group (Fig 4B and 4C). The Venn diagram describes that a total of 954 OTUs is shared among the three groups, 25 OTUs are unique to the CCO group, 26 OTUs are unique to the CCOR group, and 233 OTUs are unique to the HC group (Fig 4D).

The average composition and relative abundance of the oropharyngeal microbiome at the phylum level among the three groups were exhibited in S6A Fig and S19 Table). Difference analysis was performed at the phylum levels. The results showed that there were significant differences in the abundance of 17 phyla. With the recovery of Omicron variant infection, the

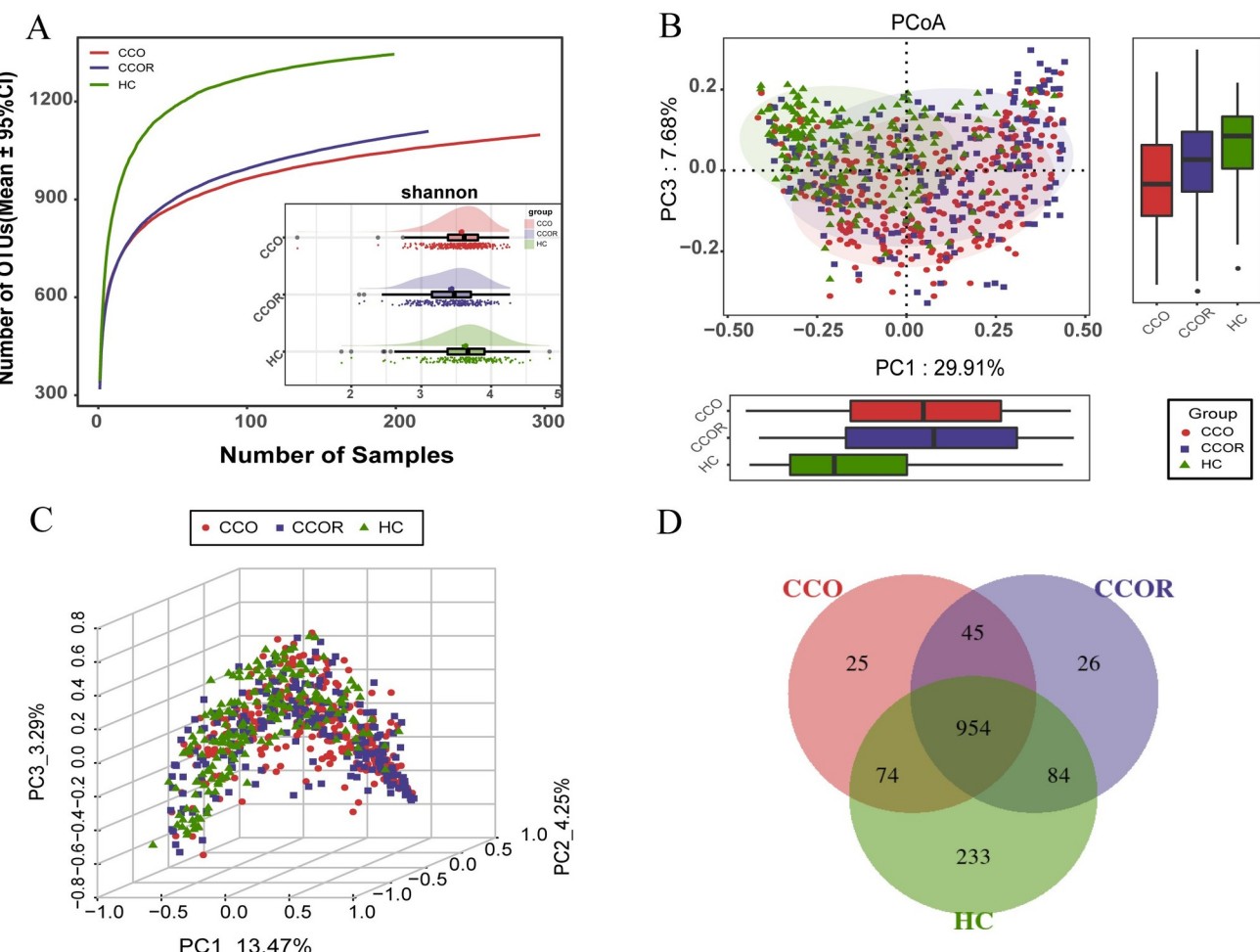

**Fig 4. The oropharyngeal microbial diversity among CCO, CCOR and HC groups.** A) Rarefaction analysis showed as the number of samples raised, the number of OTUs approached saturation in CCO (n = 297), CCOR (n = 222) and HC (n = 199) group. The number of OTUs in CCOR group and CCO group was lower than that in HC group, and the number of OTUs in CCOR group was higher than that in CCO group. Shannon index showed that the oropharyngeal microbial α-diversity in CCO and CCOR group was significantly lower than that in HC group (p<0.05). B) and C) The PCoA based on OTU distribution showed the oropharyngeal taxonomic composition was significantly different among the three groups. D) A Venn diagram displaying the overlaps among groups showed that 954 of 1441 OTUs were shared in CCO, CCOR and HC groups, while 233 of 1441 OTUs were unique to the HC group. CCO, confirmed cases of COVID-19 Omicron variant; CCOR, confirmed cases of omicron who recovered; HC, healthy controls; OTUs, operational taxonomy units; PCoA, principal coordinate analysis; NMDS, nonmetric multidimensional scaling; centerline, median; box limits, upper and lower quartiles; error bars, 95% CI.

number of *Synergistota* gradually increased (p<0.001), while the number of *Bacteria-unclassified* gradually decreased (p<0.001) (Fig 5A and S23 Table).

The average composition and relative abundance of oropharyngeal microbiomes at the genus level are shown in the S6B Fig and S21 Table. Difference analysis was performed at the genus levels. Among three groups, a total of 84 genera with significant differences in abundance (p<0.05) (Fig 5B and S24 Table). With the recovery of the infection of Omicron variant, 4 genera such as *Actinobacillus* increased gradually (p<0.05), while 7 genera such as *Atopobium* and *Lactobacillales* decreased gradually (p<0.05). In CCO group, 31 genera such as *Neisseria* and *Fusobacterium* in abundance were not different from those in CCOR group(p>0.05), but significantly decreased compared with HC group(p<0.05). There was no difference in abundance between CCO group and CCOR group of 10 genera such as *Prevotella* and

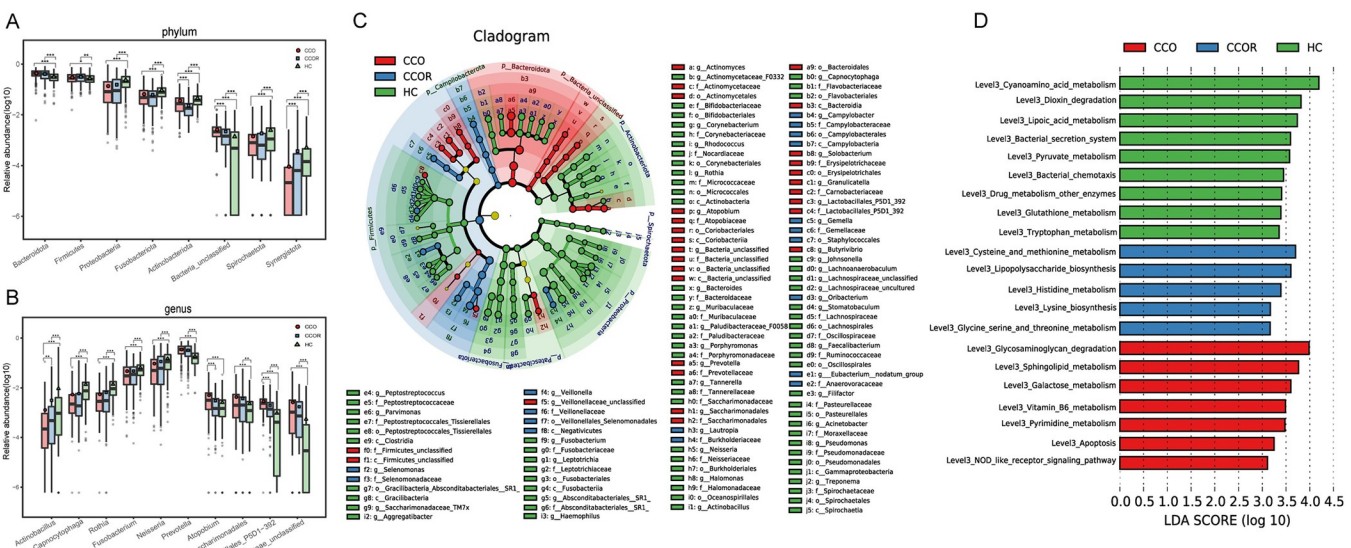

**Fig 5. Comparison of the composition and abundance of oropharyngeal microbiome among CCO (n = 297), CCOR (n = 222) and HC (n = 133) groups.**
A) In the process of infection recovery, 1 phylum gradually increased and 1 phylum decreased gradually. There was no difference between CCO group and CCOR group, but there was a significant decrease in 7 phyla compared with HC group. There was no difference between CCO group and CCOR group, but it was significantly higher in 2 phyla than that in HC group (Only part of the bacteria is displayed. Please see the S23 Table for details). B) 4 genus increased gradually during recovery of infection, while 7 genera decreased gradually. There was no difference between CCO group and CCOR group, but there was a significant decrease in 31 genera compared with HC group. There was no difference between CCO group and CCOR group, but it was significantly higher in 10 genera than that in HC group (Only part of the bacteria is displayed. Please see the S24 Table for details). C) The cladogram, representing oropharyngeal microbial structure and their predominant bacteria, revealed the greatest differences in different taxa among CCO group (n = 297), CCOR group (222) and HC group (n = 199). D) Based on the LDA selection, 37 gene functions were significantly enhanced in CCO group, 11 gene functions in CCOR group, and 41 gene functions in HC group (p<0.05, LDA>3) (Only part of the bacteria is displayed. Please see the S27 Table for details). *p<0.05, **p<0.01, ***p<0.001. CCO, confirmed cases of COVID-19 Omicron variant; CCOR, confirmed cases of omicron who recovered; HC, healthy controls; LEfSe, linear discriminant analysis (LDA) effect size. OTUs, operational taxonomy units; centerline, median; box limits, upper and lower quartiles; error bars, 95% CI.

*Saccharimonadales* (p>0.05), but there was a significant increase compared with HC group (p<0.05). The above results suggest that these bacteria may be involved in the progress or recovery of infection. We show the relative abundance changes of different OTUs in the process of infection recovery in the heatmap (S6C Fig and S25 Table).

Finally, through LEFSE analysis, we identified significant differences in abundance among three groups in the taxonomic level from phylum to genus, and the results were shown in Fig 5C and S26 Table. In addition, we found 89 enriched pathways with the most significant differences of them. Among them, 41 functions such as Cyanoamino acid metabolism, Bacterial secretion system and Pyruvate metabolism were significantly increased in HC group. 11 functions such as Lipopolysaccharide biosynthesis and threonine metabolism were significantly increased in CCOR group. Glycosaminoglycan degradation, Sphingolipid metabolism, Apoptosis and other 37 functions were significantly increased in CCO group (Fig 5D and S27 Table).

## 2.7. Oropharyngeal microbial characterization among CCO and CCOS

In order to explore whether the characteristics of oropharyngeal microbiome among different strains affect their biological behavior and clinical manifestations, we included pharyngeal samples from 73 patients infected with original strains (The patient information of the original strain can be seen in the previously published article [20]). In order to avoid the excessive difference in sample size between the two groups, we used the PSM to control gender, age and other confounding factors, 134 patients were selected from the CCO group for further analysis.

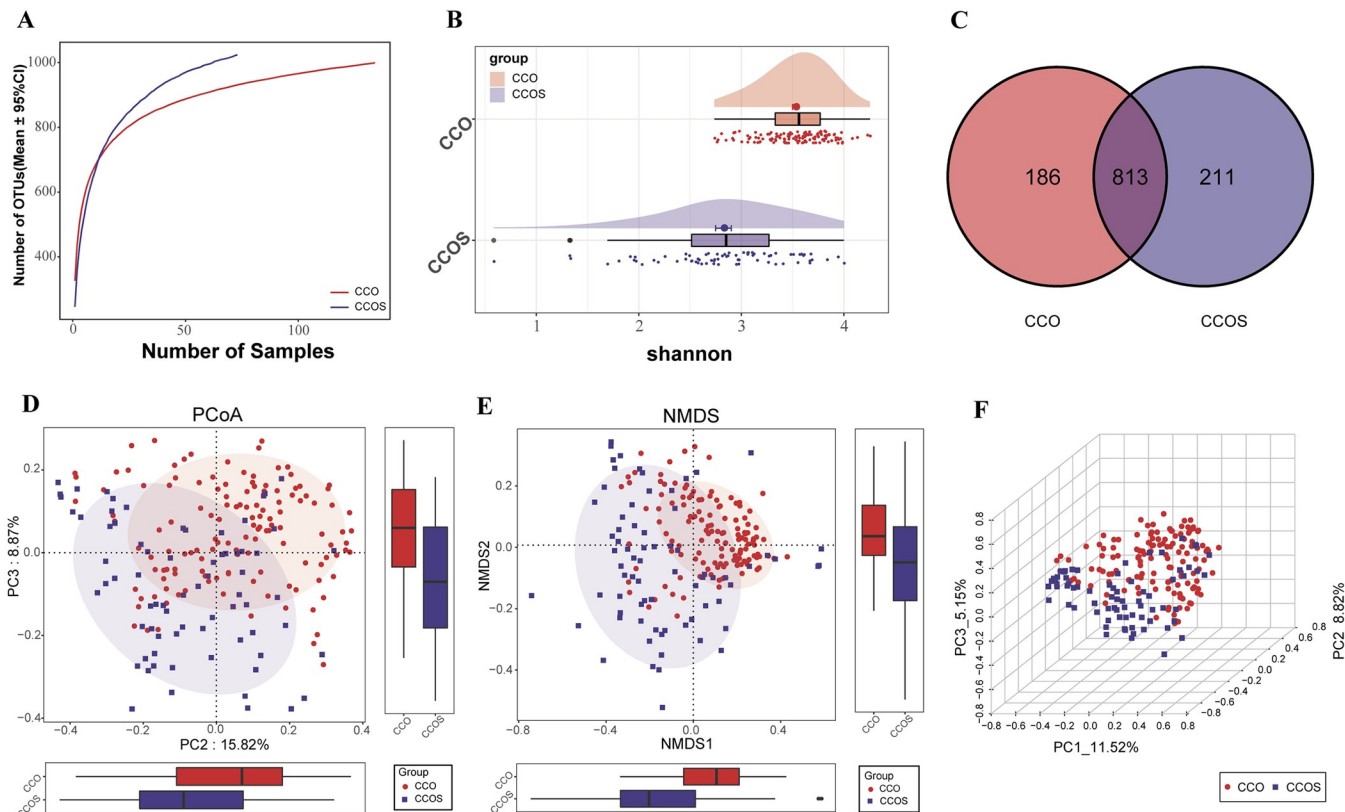

**Fig 6. The oropharyngeal microbial diversity between CCO (n = 134) and CCOS (n = 73) groups.** A) Rarefaction analysis showed as the number of samples raised, the number of OTUs approached saturation in CCO (n = 134) and CCOS (n = 73). B) Shannon index showed that the oropharyngeal microbial α-diversity in CCOS group was significantly lower than that in CCO group (p<0.001). C) A Venn diagram displaying the overlaps between groups showed that 813 of 1210 OTUs were shared in CCO and CCOS groups, while 211 of 1210 OTUs were unique to the CCOS group. D) and F) The PCoA and E) NMDS based on OTU distribution showed the oropharyngeal taxonomic composition was significantly different between the two groups. CCO, confirmed cases of COVID-19 Omicron variant; CCOS, confirmed cases of original strain; OTUs, operational taxonomy units; PCoA, principal coordinate analysis; NMDS, nonmetric multidimensional scaling; centerline, median; box limits, upper and lower quartiles; error bars, 95% CI.

The results of rarefaction analysis show that the OTUs abundance of CCO group and CCOS group was close to saturation (Fig 6A). The results of Shannon index showed that the oropharyngeal microbial alpha diversity in CCO group was significantly higher than that in CCOS group (p<0.001) (Fig 6B and S28 Table). The results of NMDS and PCoA analysis showed that there was a significant difference in the distribution of oropharyngeal microbial community between two groups (p<0.001) (Fig 6D–6F). The Venn diagram showed that 813 OTUs were shared between the CCO group and the CCOS group, 186 OTUs were unique to the CCO group and 211 OTUs were unique to the CCOS group (Fig 6C).

Then, we further identified the composition and changes of oropharyngeal microbiomes between two groups. At the phylum level, the abundance of the main phyla in CCO group from high to low was in the order of *Bacteroidota*, *Firmicutes*, *Proteobacteria* and *Fusobacteriota*, and its sum accounted for 93.75% of the total microbiota abundance. In CCOS group, the abundance of the main phyla from high to low was in the order of *Firmicutes*, *Bacteroidota*, *Proteobacteria* and *Fusobacteriota*, and its sum accounted for 88.97% of the total microbiota abundance (Fig 7A and S29 Table). The results of Wilcoxon rank-sum test show that a total of 10 phyla with significant differences in abundance were identified (p<0.05) (Fig 7B and S33 Table). Between them, 6 phyla such as *Bacteroidota* and *Patescibacteria* were increased in CCO

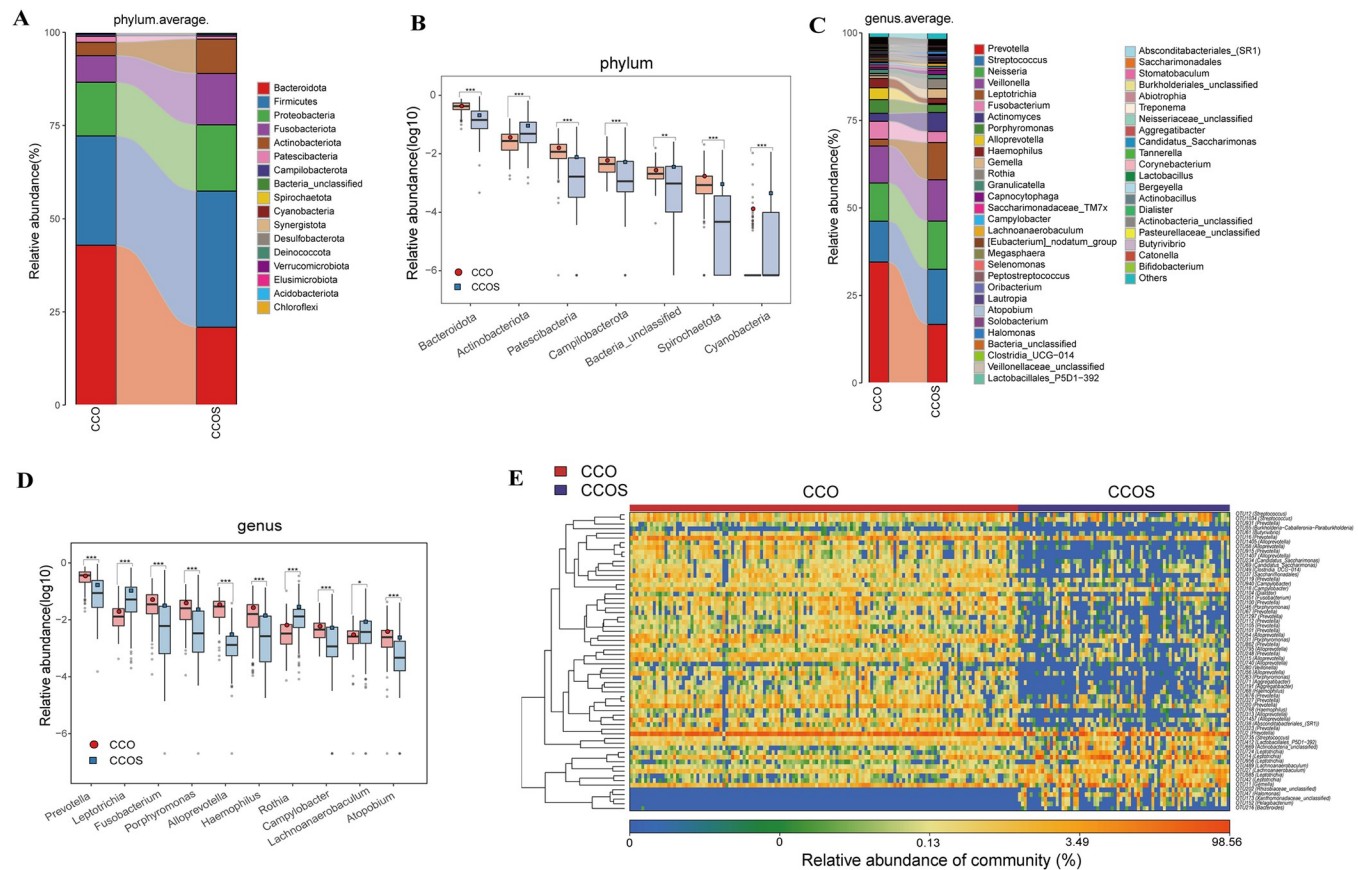

**Fig 7. Comparison of the composition and abundance of oropharyngeal microbiome between CCO (n = 134) and CCOS (n = 73) groups.** A) Average compositions and relative abundance of the bacterial community at the phylum level between CCO and CCOS groups. B) Compared with CCOS group, 6 phyla increased significantly and 4 phyla decreased significantly in CCO group (Only part of the bacteria is displayed. Please see the S33 Table for details). C) Average compositions and relative abundance of the bacterial community at the genus level between CCO and CCOS groups. D) Compared with CCOS group, 23 genera increased significantly and 19 genera decreased significantly in CCO group (Only part of the bacteria is displayed. Please see the S34 Table for details). E) Heatmap showed the relative abundances of differential OTUs for each sample between CCO and CCOS groups. *p < 0.05, **p < 0.01, ***p < 0.001. CCO, confirmed cases of COVID-19 Omicron variant; CCOS, confirmed cases of original strain; OTUs, operational taxonomy units; centerline, median; box limits, upper and lower quartiles; error bars, 95% CI.

group (p<0.05), while 4 phyla such as *Cyanobacteria* and *Actinobacteriota* were increased in CCOS group (p<0.05).

The average composition and relative abundance of oropharyngeal microbiomes at the genus level are shown in the Fig 7C and S31 Table. In CCO group, the abundance of the main genera from high to low was in the order of *Prevotella*, *Streptococcus*, *Neisseria*, *Veillonella* and *Fusobacterium*, its sum accounted for 72.79% of the total microbiota abundance. In CCOS group, the abundance of the main genera from high to low was in the order of *Prevotella*, *Streptococcus*, *Neisseria*, *Veillonella* and *Leptotrichia*, its sum accounted for 68.67% of the total microbiota abundance. A total of 42 genera with significant differences in abundance were identified by Wilcoxon rank-sum test (p<0.05) (Fig 7D and S34 Table). Between them, 23 genera such as *Prevotella*, *Fusobacterium* and *Alloprevotella* in CCO group were increased compared with CCOS group (p<0.05), while 19 genera such as *Leptotrichia*, *Rothia* and *Lachnoanaerobaculum* were increased in CCOS group (p<0.05). We selected 44 OTUs with significant differences between two groups, and the abundance content was represented by color in the heatmap (Fig 7E and S35 Table). To sum up, our results revealed the differences in the composition of

oropharyngeal microbiomes between infection with original strain and Omicron variant. In CCO group, it is characterized by the increase of gram-negative anaerobes such as *Prevotella*, *Fusobacterium*, which are conditional pathogens, some of which can induce inflammation, change intestinal permeability and produce lipopolysaccharide, and other substances. In CCOS group, it was characterized by an increase in both Gram-negative and Gram-positive bacteria, such as *Leptotrichia* and *Rothia*. They are also conditioned pathogens and often play a role in mixed infections, disturbing the physiological and immune balance of the body.

Finally, through LEfSe analysis, we identified a significant difference in abundance between two groups in the taxonomic level from phylum to genus, which was shown in the evolutionary cladogram (S7A Fig and S36 Table). In addition, 74 enriched pathways with the most significant differences between two groups were found. In the CCO group, 41 functions such as Plant pathogen interaction, Apoptosis and NOD like receptor signaling pathway were significantly increased. In the CCOS group, 33 functions such as Biosynthesis of ansamycins, Pyruvate metabolism and Bacterial secretion system were significantly increased (S7B Fig and S37 Table).

## 2.8. Correlation between the Oropharyngeal Microbiota and Clinical Indicators

In order to research the association between the changes of oropharyngeal microbiota and the outcome of the disease, we further analyzed the correlation between 27 oropharyngeal OTUs and 6 clinical indicators in CCO group and HC group (Fig 8 and S38 Table). The results showed that four clinical indexes (white blood cell, eosinophils, platelets and urea) were closely related to less than 6 OTUs. Among them, white blood cell is negatively correlated with OTU63 (*Butyrivibrio*) and positively correlated with OTU28 (*Rothia*) and other two OTUs. Platelet was negatively correlated with OTU360 (*Prevotella*) and positively correlated with OTU28 (*Rothia*) and other three OTUs. Urea was negatively correlated with OTU588 (*Prevotella*) and positively correlated with OTU78 (*Johnsonella*) and OTU48 (*Campylobacter*). The other two indexes (neutrophils, albumin) were closely related to more than 6 OTUs. Among them, neutrophils were negatively correlated with 6 OTUs (including *Prevotella*, *Butyrivibrio*, etc.) and positively correlated with 5 OTUs (including *Johnsonella*, *Rothia*, etc.). Albumin were negatively correlated with 12 OTUs (including *Veillonella*, *Prevotella*, etc.) and positively correlated with 15 OTUs (including *Johnsonella*, *Rhodococcus*, etc.). In summary, changes in oropharyngeal microbiota may affect the severity and outcome of the disease.

## 3. Discussion

Based on a large sample size, our research characterized the pharyngeal microbiome of COVID-19 Omicron strain infection. We found that the patients infected with Omicron variant had microbiomes imbalance in oropharynx, and the microbial diversity was significantly less than that in healthy individuals. The predominant bacterial composition in the Omicron variant at the genus level included *Prevotella*, *streptococcus*, *Neisseria* and *Veillonella*, this result is similar to the research of Ma et al. [21]. Because of the different strains, there are some differences in the composition and abundance of oropharyngeal microbiomes. Compared with HC group, some conditional pathogens such as *Prevotella* and *Veillonella* were significantly increased in CCO group, while some acidogenic bacteria such as *Fusobacterium* and *Alloprevotella*, which regulate metabolic function, were significantly decreased in CCO group. These unbalanced microbiomes can directly affect and aggravate the infection of novel coronavirus [22, 23], and also cause clinical symptoms by destroying the immune balance of the host and causing secondary infection [24, 25]. The lipopolysaccharide contained in *Prevotella* can activate epithelial cells to produce IL-8, IL-6 and CCL20, thus promoting mucosal Th17 immune

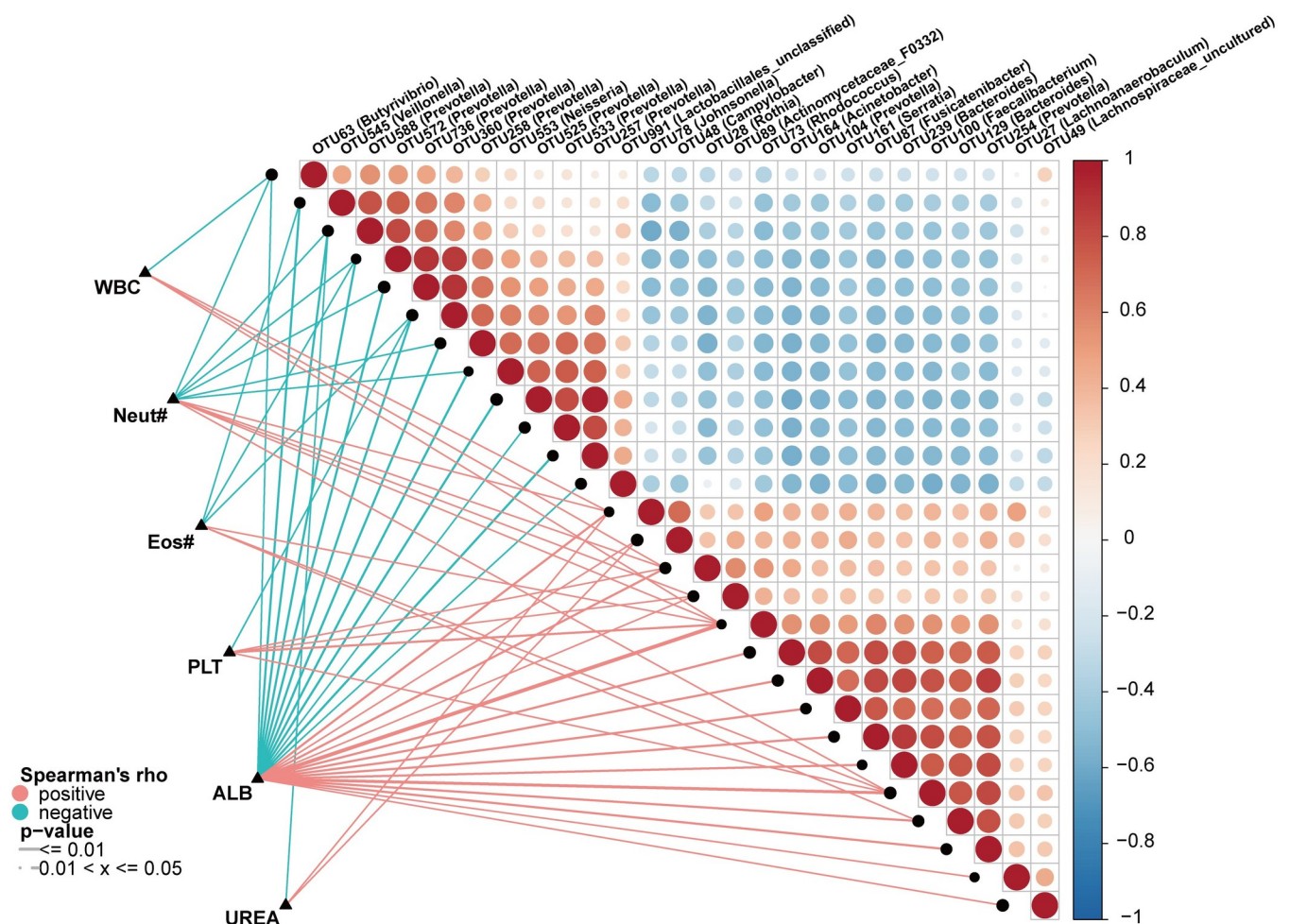

**Fig 8. Associations between the oropharyngeal microbiomes and clinical indices of Omicron variant.** The Heatmap shows the Spearman's correlation coefficients between 27 oropharyngeal OTUs and 6 clinical indicators in CCO (n = 297) and HC (n = 199) groups. The red line indicates a positive correlation, and the blue line indicates a negative correlation. OTUs, operational taxonomic units; CCO, confirmed cases of COVID-19 Omicron variant; HC, healthy controls; WBC, white blood cell; Neut, neutrophils; Eos, eosinophils; PLT, platelets; ALB, albumin; CREA, creatinine.

response and neutrophil recruitment. Therefore, although *Prevotella* is a resident bacterium, its enrichment may lead to a cytokine storm [26]. And it also closely related to localized infection of ear and nose [27,28], inflammatory periodontitis [29,30], pulmonary infection [31,32], gastrointestinal infection [33,34] and some autoimmune diseases. Moreover, immunological interactions between *Prevotella* and viruses usually inducing an increase of pro-inflammatory responses [35]. Virulence factors including lipopolysaccharide [36,37], fimbriae [38], hemolysin [39] and adhesin [40] were found in *Prevotella*, which can promote its pathogenicity and survival in the host. These studies show that *prevotella* not only promotes the growth of SARS-CoV-2, but also interacts with certain mediators of disease progression. *Veillonella* is a common bacterium in oropharynx, respiratory tract and digestive tract. It is an important pathogen of periodontitis and probably plays a role in a variety of mixed infections. Studies have shown that the combination of *Veillonella* and *Streptococcus* can counteract the production of IL-12p70 and enhance the response of IL-8, IL-6, IL-10 and TNF- α, which may be related to the cytokine storm after COVID-19 infection. Ma et al. reported that *Veillonella* was the most prominent biomarker for COVID-19 compared with flu patients or healthy controls, and it

also one of the bacteria to predict the severity of COVID-19 [21]. Although the types of strains in research are different, it shows that microbiome dysbiosis can promote the inflammatory environment favoring coronavirus invasion and viral replication [41]. *Neisseria*, *Fusobacterium* and *alloprevotella*, which are also natural components of oropharyngeal microbes, may play a physiological role in regulating the colonization of potential pathogens. Their metabolites are mainly short-chain fatty acids, such as butyric acid, propionic acid and so on. Several pieces of evidence show that short-chain fatty acids have the effects of anti-inflammation, anti-oxidation, inhibiting cancer and promoting the integrity of intestinal epithelial barrier function [42–44]. Most importantly, butyric acid can not only enhance the antibacterial function of macrophages [45], but also significantly inhibit the production of proinflammatory cytokines, chemokines and calprotectin by neutrophils, and reduce the release of MPO, ROS and NETs (neutrophil extracellular traps) formation [46]. NETs are released by neutrophils, which can kill microorganisms. However, excessive generation of NETs will lead to many negative effects. When neutrophils are exposed to live SARS-CoV-2 virus, they are more likely to form NETs than other neutrophils [47]. NET plays a key role in cytokine storm and thromboregulatory disorder after infection with novel coronavirus [48]. NET production also has been considered as a predictor to assess disease severity [49] and clinical outcomes during infection [50]. Therefore, the decrease of these acidogenic bacteria may lead to aggravation of pulmonary infection, destruction of intestinal epithelial barrier and even increase the probability of "cytokine storm". In addition, we constructed a diagnostic model based on 6 optimal oropharyngeal OTUs in the discovery cohort (198 CCO and 133 HC), and achieved extremely high diagnostic performance in the verification cohort (99 CCO and 66 HC), with an AUC value of 0.984. In order to verify the universality of the diagnostic model, we included a set of cross-age verification cohorts. The results show that it has high diagnostic efficiency for middle-aged and elderly patients with infection. In order to verify the specificity of the diagnostic model, we included another subtype of Omicron variant XBB.1.5, and found that this diagnostic model can still accurately distinguish between BA.5.2 and XBB.1.5. These findings suggest a novel non-invasive tool for clinically diagnosing Omicron variant BA.5.2 infections.

We found that the overall microbial community composition of mild infection and moderate infection changed. A significant increase in *Streptobacillus*, *Lachnoanaerobaculum* and *Eubacterium Nodatum* was observed in the moderate infection. These results suggest that the change of oropharyngeal microbiomes may affect the severity of the disease, and the changed microbes are expected to become the predictor of the severity of infection of COVID-19 Omicron variant. Under the condition of excluding a variety of confounding factors, we analyzed the differential expression of bacterial species between the CCO2 group and CCO3 group. The results show that enrichment of *Granulicatella*, *lactobacillales* and *corynebacterium* in CCO2 compared with CCO3 (p<0.05). These results suggest that the oropharyngeal microbiota may be related to the efficacy of COVID-19 vaccine, and oropharyngeal dysbiosis may reduce the COVID-19 vaccine efficacy and persistence. We reported the characteristics of oropharyngeal microbiomes of confirmed cases of Omicron who recovered. In CCOR group, some conditional pathogens such as *Prevotella* and *Veillonella* in abundance were not different from that in CCO group, but significantly increased compared with HC group. There was no difference between CCOR group and CCO group in *Neisseria*, *Fusobacterium* and some other acidogenic bacteria, but significantly decreased compared with HC group. The above results suggest that these bacteria may play a role in the development of the disease. On the other hand, it may be related to the post COVID-19 conditions. It should be noted that *Actinobacillus* gradually increased in the process of infection recovery, while *Atopobium*, *Lactobacillales* and some other lactic acid-producing bacteria gradually decreased in the process of infection recovery. Lactic acid is the product of anaerobic metabolism of glucose in the body. Excessive lactic acid

may be neurotoxic or lead to coagulation dysfunction [51]. In the immune system, lactic acid can affect the cytotoxic activity of NK and induce its apoptosis by reducing the intracellular PH value [52]. At the same time, it can also interfere with the secretion of TNF and glycolysis of monocytes, inhibit the differentiation of monocytes to DCs, and inactivate the cytokines secreted by differentiated DCs [53]. Lactic acid can also damage the cytotoxicity of T cells, especially CD8+T cells in vitro. Research shows that lactic acid can inhibit the proliferation of 95% of human cytotoxic T lymphocytes and the production of related cytokines such as IL-2 and IFN- γ, and inhibit 50% of the cytotoxic activity [54]. The decrease of lactic acid not only promote the recovery of immune system function, but also represent the normal oxygen supply and blood coagulation function, which is conducive to the recovery of infection. Some studies have shown that *Atopobium* can significantly promote the expression of chemokines such as IL-8, MIP-3α in cervical epithelial cells, which is a key trigger of inflammation and vaginal epithelial innate immune response [55,56]. Although the relationship between *Atopobium* and pulmonary infection has not been fully studied, it provides us with a new idea.

In this study, we compared and analyzed the oropharyngeal microbiomes of Omicron variant and the original strain. The results showed that *Prevotella*, *Fusobacterium* and some other Gram-negative anaerobes increased significantly in the CCO group, and the endotoxin (also known as lipopolysaccharide) produced by them could activate complement, coagulation and kinin system [57]. TLR4 is the main endotoxin receptor [58]. After activation, it can stimulate monocytes, phagocytes and other cells to release cytokines such as TNF-α, IL-1b and IL-10, thus mediating lung injury and even multiple organ injury [59]. In CCOS group, *Leptotrichia*, *Rothia*, *Lachnoanaerobaculum* and other bacteria increased significantly, including both Gram-negative bacteria and Gram-positive bacteria. *Rothia* is currently considered to be an opportunistic pathogen that mainly affects immunocompromised hosts. As a cause of endocarditis after cardiac catheterization, it has been identified as a human pathogen for the first time [60]. It is now more related to severe infectious diseases such as pneumonia, acute bronchitis, meningitis, peritonitis, cancer, severe neutropenia, diabetes and other severe immune impaired diseases [61]. It is reported that it may cause pneumonia not only in patients with low immune function, but also in patients with chronic lung diseases with normal immune function. We found that both the Omicron variant and the original strain have their own unique characteristics of oropharyngeal microbiomes, and whether it is associated with different clinical outcome between them needs to be further studied. However, this provides clues for early diagnosis and disease treatment among different strains of novel coronavirus. Early and timely detection of pathogens and appropriate antibiotic treatment is an important condition for disease recovery. The administration of specific probiotics may also represent a new intervention program for COVID-19 patients.

Clinical index is one of the important links for us to judge the severity of the disease. Therefore, we studied the relationship between clinical indexes and oropharyngeal microbiota. We found that neutrophils were negatively correlated with OTU588 (*Prevotella*) (p<0.001, Rho = -0.23) and *Prevotella* was significantly increased in CCO group. Lipopolysaccharide produced by *Prevotella* can induce phagocytes to release pro-inflammatory cytokines such as IL-6, TNF-α and IL-1β. The disorder of pro-inflammatory cytokine environment may interfere with the migration and transport of leukocytes [62]. Therefore, we think that the increase of *Prevotella* after infection with COVID-19 Omicron variant may promote the occurrence of secondary infection by inhibiting neutrophils, thus leading to further aggravation of the disease.

Our research shows that Omicron variant infections may disrupt oropharyngeal microbiota, exacerbating disease severity possibly through the secretion of endotoxins. These discoveries offer new insights for targeting oropharyngeal microbiota to alleviate disease progression or prevent variant strain infections. Although the sample size included in our study is very

large, due to the limitations of the actual situation, most of the samples we collected are mild or asymptomatic patients, and moderate patients account for only a small part of them. There are no severe patients and critical patients. Therefore, there are some limitations in exploring the relationship between oropharyngeal microbiomes and the severity of the disease. In addition, due to the limited laboratory conditions, it is not possible to establish animal models to verify our results. However, with the further research, the use of oropharyngeal microbial markers may play a unique role in the diagnosis, treatment and prevention of the disease.

## 4. Materials and methods

### Ethics Approval and consent to participate

This study was reviewed and approved by the Institutional Review Board from The First Affiliated Hospital of Zhengzhou University (L2021-Y429-002). All participants signed written informed consent form after learning about the research scheme.

### 4.1. Study design

The design of this study is based on the principles of the prospective specimen collection and retrospective blind evaluation, and under the guidance of the Helsinki Declaration and the Rules of Good Clinical Practice.

We prospectively collected throat swab samples from some inpatients in COVID-19 designated hospital in Henan Province from January to February 2022. All patients were confirmed to be infected with Omicron variant BA.5.2 branch. The diagnosis is based on the "COVID-19 diagnosis and treatment Program Trial V.9 Guidelines" issued by the National Health Commission of the people's Republic of China (S1 Method). After rigorous screening, we included 297 confirmed cases of COVID-19 Omicron variant, 199 healthy controls and 73 confirmed cases of original strain. In March 2022, we collected some middle-aged and elderly patients infected with Omicron variant BA.5.2 branch in the same designated hospital, and 20 of them were selected as the cross-age verification cohort of this study. All patients were treated with standard guidelines. When confirmed patients meets the discharge criteria in the guidelines for diagnosis and treatment, we included throat swab samples from 222 confirmed cases of omicron who recovered. Throat swab samples of 80 cases of Omicron variant subtype XBB.1.5 patients were collected in COVID-19 designated Hospital of Henan Province from April to May 2023. Finally, all pharyngeal swab samples were sequenced and analyzed by 16S rRNA Miseq. The healthy control samples came from the volunteers in the physical examination center of the First Affiliated Hospital of Zhengzhou University (S1 Table). Details of inclusion, exclusion, and diagnostic criteria can be found in S1 Method.

### 4.2. Oropharyngeal Specimen Collection and DNA Extraction

In order to ensure a sufficient effective sample size, two sets of oropharyngeal samples were collected from each subject. In order to avoid factors such as diet and drugs interfering with oral microecological changes as much as possible, we asked each subject to have an empty stomach for at least 2 hours and gargle with normal saline for 2–3 times before collecting the samples. Specially trained operators scrape the posterior pharyngeal wall and both sides of the tonsil with a pharyngeal swab 2–3 times, and then immerse the head of the pharynx swab into a test tube containing 2-3ml virus preservation solution. All the samples were inactivated in a warm water bath at 56°C for more than 30 minutes, and finally stored in the freezer at -80°C.

We use a Qiagen Mini Kit (Qiagen, Hilden, Germany) to extract DNA from oropharyngeal microorganisms, and the operation procedure follows the manufacturer's instructions. The

DNA samples were quantified by a Qubit 2.0 Fluorometer (Invitrogen, Carlsbad, CA, USA) and molecular size was determined by agarose gel electrophoresis. Finally, the microbial samples were diluted to 10 ng μ L-1 for further analysis.

### 4.3. PCR Amplification and 16S rRNA Gene Sequencing

We continue to use the previous method for PCR amplification and DNA library construction. ABI GeneAmp 9700 (Thermo Fisher Scientific, Waltham, MA, USA) was used for polymerase chain reaction, and the products were mixed to construct a DNA library. All samples were sequenced using the Illumina MiSeq platform in Shanghai Mobio Biomedical Technology, China. All raw Illumina readings are stored in the European Bioinformatics Institute European Nucleotide Archives Database (PRJNA911036). The results of the amplification are processed by us, and the details are displayed in S1 Method.

### 4.4. Operational Taxonomy Unit (OTU) Clustering and taxonomy annotation

We use abundance to obtain Quantity-controlled sequences from all samples and bin the OTUs with the UPARSE pipeline, then identify the representative sequence. Set 97% as the homology threshold, gather the gene sequence into OTUs, and discard the OTUs with low coverage. The total OTUs at different levels including phylum, class, order, family and genus was counted, and the phylogenetic relationship of each OTU was marked by RDP classifier V.2.6 according to the developer's documents.

### 4.5. Bioinformatic Analysis of 16S rRNA Sequencing

The sample size and sequencing depth are determined to be saturated by rarefaction curves and species accumulation curves. bacterial α-diversity was analyzed by using the R program package "vegan" and expressed by Ace index, Chao index and Shannon index and Simpson index. The "R package" (http://www.R-project.org/) visualizes the microbial space between samples by generating principal coordinates analysis (PCoA) and non-metric multidimensional scaling (NMDS). The Venn diagram is used to reveal the similarity and overlap of OTUs among different groups, and to identify common and unique OTUs in multiple groups. The Heatmap Builder draws a heatmap describing the key variables.

Linear discriminant analysis (LDA) effect size (LEfSe) method (http://huttenhower.sph.harvard.edu/lefse/)e/) was used to identify the key microflora with significant differences (Kruskal-Wallis rank-sum test, p<0.05), and then linear discriminant analysis (LDA) was used to analyze the differences at the taxonomic level. The cutoff value was set as LDA score (log 10) = 2.5 or 3. The evolutionary cladistic diagram depicts the taxonomic analysis of bacteria at different levels (including phylum, class, order, family and genus). Spearman correlation analysis was conducted to study the correlation between clinical indexes and oropharyngeal microflora.

The diagnostic model was constructed through 5 times of fivefold cross-verification (R3.4.1, random forest 4.6–12 packages), and then the cross-validation error curve was found. The point with the minimum cross-validation error was viewed as the cut-off point, then listed all sets of OTUs markers that are less than the cut-off value, and select the set with the least number of OTUs as the best set. POD (possibility of disease) index was defined as the ratio between the number of randomly generated decision trees that predicted sample as "CCO" and that of "HC", It can indicate the possibility of diagnosis of the disease. The receiver operating characteristic (ROC) curve was constructed with the "R3.3.0 pROC package". If the area under the ROC curve (AUC) is greater than 0.7, it is considered that it is feasible to use microbial markers as a tool for disease diagnosis (S1 Method).

### 4.6. Statistical analysis

Using SPSS v.20.0 for Windows (SPSS, Chicago, Illinois, USA) to analyze the data. Comparing the differences between the two groups, we use t-test to analyze continuous variables of normal distribution; use Mann Whitney U test to analyze continuous variables of non-normal distribution; and use chi-square test to analyze classified variables. To compare the differences among the three groups, we use one-way analysis of variance to compare the continuous variables of normal distribution and Kruskal-Wallis test to compare the continuous variables of non-normal distribution. $p < 0.05$ (two-tailed) was defined as statistical significance.

## Supporting information

**S1 Fig. Research design.** A total of 835 oropharyngeal specimens were collected from Henan Province, China. After screening, 791 samples were sequenced by 16S rRNA MiSeq, including 297 CCO, 222 CCOR, 199 HC and 73 CCOS. CCO, confirmed cases of COVID-19 Omicron variant; CCO2, patients in CCO group who received two times of the vaccine; CCO3, patients in CCO group who received three times of the vaccine; CCOR, confirmed cases of COVID-19 Omicron who recovered; HC, healthy controls; CCOS, confirmed cases of COVID-19 Original strain. RFC, random forest classifier.
(JPG)

**S2 Fig. Clinical characteristics of subjects in discovery and validation cohorts.** We presented continuous variables as the means (standard deviations) and categorical variables as percentages. Differences between subjects with SARS-CoV-2 Omicron strain infection (n = 198, n = 99) and healthy controls (n = 133, n = 66) were carried out by using Student's t-test for normally distributed continuous variables, the Wilcoxon rank-sum test for non-normally distributed continuous variables, and the chi-square test for categorical variables. Statistical significance was defined by $p < 0.05$ (two-tailed).
(JPG)

**S3 Fig. LEfSe analysis between CCO and HC.** A) The cladogram, representing oropharyngeal microbial structure and their predominant bacteria, revealed the greatest differences in different taxa between CCO group (n = 198) and HC group (n = 133). B) Based on the LDA selection, 47 gene functions were significantly enhanced in CCO group and 42 gene functions in HC group. ($p < 0.05$, LDA>3). CCO, confirmed cases of Omicron variant; HC, healthy controls; LEfSe, linear discriminant analysis (LDA) effect size (Only part of the bacteria is displayed. Please see the S12 Table for details).
(JPG)

**S4 Fig. Diagnostic efficacy of oropharyngeal microbial classifier on Omicron variant.** A) The POD value was significantly higher in CCO' group (n = 20) compared with that in HC' group (n = 20) in the verification cohort. B) The POD value achieved an AUC of 98.42% (95% CI 95.14% to100%, $p < 0.0001$) between CCO' group (n = 20) versus HC' group (n = 20) in the verification cohort. C) The POD value was significantly higher in CCO group (n = 99) compared with that in CCXBB group (n = 80) in the verification cohort. D) The POD value achieved an AUC of 90.59% (95% CI 86.23% to94.94%, $p < 0.0001$) between CCO group (n = 99) versus CCXBB group (n = 80) in the verification cohort. CCO, confirmed cases of COVID-19 Omicron variant; HC, healthy controls; CCO', Cross-age verification cohort of COVID-19 Omicron variant; HC', Cross-age verification cohort of healthy controls; CCXXB, confirmed cases of Omicron subvariants XBB.1.5; OTUs, operational taxonomy units; POD, probability of disease; AUC, area under the curve; centerline, median; box limits, upper and

lower quartiles; error bars, 95% CI.
(JPG)

**S5 Fig. Subgroup analysis of infection severity and vaccination times of Omicron variant.**
A) Shannon index showed that there was no significant difference in microbial α-diversity between moderate and mild patients. (p>0.05). B) Simpson index showed that there was no significant difference in microbial α-diversity between moderate and mild patients. (p>0.05). C) A Venn diagram displaying the overlaps between groups showed that 712 of 934 OTUs were shared in Mild and Moderate groups, while 31 of 934 OTUs were unique to the Moderate group. D) The PCoA based on OTU distribution showed the oropharyngeal taxonomic composition was significantly different between the two groups. E) Compared with Mild group, 3 genera increased significantly in Moderate group. F) Compared with CCO2 group, 3 genera decreased significantly in CCO3 group. *p<0.05, **p<0.01, ***p<0.001. CCO, confirmed cases of COVID-19 Omicron variant; HC, healthy controls; CCO2, patients in CCO group who received two times of the vaccine; CCO3, patients in CCO group who received three times of the vaccine; OTUs, operational taxonomy units; PCoA, principal coordinate analysis; centerline, median; box limits, upper and lower quartiles; error bars, 95% CI.
(JPG)

**S6 Fig. Comparison of the composition and abundance of oropharyngeal microbiome among CCO (n = 297), CCOR (n = 222) and HC (n = 133) groups.** A) Average compositions and relative abundance of the bacterial community at the phylum level among CCO, CCOR and HC groups. B) Average compositions and relative abundance of the bacterial community at the genus level among CCO, CCOR and HC groups. C) Heatmap showed the relative abundances of differential OTUs for each sample among CCO, CCOR and HC groups. CCO, confirmed cases of COVID-19 Omicron variant; CCOR, confirmed cases of omicron who recovered; HC, healthy controls; OTUs, operational taxonomy units; centerline, median; box limits, upper and lower quartiles; circle or square symbol, mean; error bars, 95% CI.
(JPG)

**S7 Fig. LEfSe analysis between CCO and CCOS.** A) The cladogram, representing oropharyngeal microbial structure and their predominant bacteria, revealed the greatest differences in different taxa between CCO group (n = 134) and CCOS group (n = 73). B) Based on the LDA selection, 41 gene functions were significantly enhanced in CCO group and 33 gene functions in CCOS group. (p<0.05, LDA>3). CCO, confirmed cases of Omicron variant; CCOS, confirmed cases of original strain; LEfSe, linear discriminant analysis (LDA) effect size (Only part of the bacteria is displayed. Please see the S37 Table for details).
(JPG)

**S1 Table. Detailed clinical data. Including confirmed cases of COVID-19 Omicron variant (CCO = 297).**
(XLSX)

**S2 Table. Detailed clinical data. Including healthy controls (HC = 199).**
(XLSX)

**S3 Table. The detailed values of oropharyngeal microbial α-diversity index and observed OTUs in the discovery cohort (CCO = 198, HC = 133).**
(XLSX)

**S4 Table. The relative abundance and average composition at the phylum level in discovery cohort (CCO = 198, HC = 133).**
(XLSX)

**S5 Table. The abundance and composition at the phylum level of all samples in discovery cohort (CCO = 198, HC = 133).**
(XLSX)

**S6 Table. The relative abundance and average composition at the genus level in discovery cohort (CCO = 198, HC = 133).**
(XLSX)

**S7 Table. The abundance and composition at the genus level of all samples in discovery cohort (CCO = 198, HC = 133).**
(XLSX)

**S8 Table. The different degree of phylum level (p value) in discovery cohort (CCO = 198, HC = 133).**
(XLSX)

**S9 Table. The different degree of genus level (p value) in discovery cohort (CCO = 198, HC = 133).**
(XLSX)

**S10 Table. The relative abundance and distribution of the key 59 OTUs in discovery cohort (CCO = 198, HC = 133).**
(XLSX)

**S11 Table. The cladogram of oropharyngeal microbial structure and their predominant bacteria in discovery cohort (CCO = 198, HC = 133).**
(XLSX)

**S12 Table. The corresponding LDA value and p value of microbial community gene function for samples in the discovery cohort (CCO = 198, HC = 133).**
(XLSX)

**S13 Table. The corresponding output value and POD value of each optimal microbial marker in the discovery cohort (CCO = 198, HC = 133).**
(XLSX)

**S14 Table. The corresponding output value and POD value of each optimal microbial marker in the validation cohort (CCO = 99, HC = 66).**
(XLSX)

**S15 Table. The detailed values of oropharyngeal microbial α-diversity index and observed OTUs in mild and moderate patients (mild = 57, moderate = 19).**
(XLSX)

**S16 Table. The different degree of genus level (p value) in mild and moderate patients (mild = 57, moderate = 19).**
(XLSX)

**S17 Table. The different degree of genus level (p value) in CCO2 and CCO3 groups (CCO2 = 60, CCO3 = 20).**
(XLSX)

**S18 Table. The detailed values of oropharyngeal microbial α-diversity index and observed OTUs among groups CCO (n = 297), CCOR (n = 222) and HC (n = 199).**
(XLSX)

**S19 Table. The relative abundance and average composition at the phylum level among groups CCO (n = 297), CCOR (n = 222) and HC (n = 199).**
(XLSX)

**S20 Table. The abundance and composition at the phylum level of all samples among groups CCO (n = 297), CCOR (n = 222) and HC (n = 199).**
(XLSX)

**S21 Table. The relative abundance and average composition at the genus level among groups CCO (n = 297), CCOR (n = 222) and HC (n = 199).**
(XLSX)

**S22 Table. The abundance and composition at the genus level of all samples among groups CCO (n = 297), CCOR (n = 222) and HC (n = 199).**
(XLSX)

**S23 Table. The different degree of phylum level (p value) among groups CCO (n = 297), CCOR (n = 222) and HC (n = 199).**
(XLSX)

**S24 Table. The different degree of genus level (p value) among groups CCO (n = 297), CCOR (n = 222) and HC (n = 199).**
(XLSX)

**S25 Table. The relative abundance and distribution of the key 69 OTUs among groups CCO (n = 297), CCOR (n = 222) and HC (n = 199).**
(XLSX)

**S26 Table. The cladogram of oropharyngeal microbial structure and their predominant bacteria among groups CCO (n = 297), CCOR (n = 222) and HC (n = 199).**
(XLSX)

**S27 Table. The corresponding LDA value and p value of microbial community gene function for samples in groups CCO (n = 297), CCOR (n = 222) and HC (n = 199).**
(XLSX)

**S28 Table. The detailed values of oropharyngeal microbial α-diversity index and observed OTUs between groups CCO (n = 134) and CCOS (n = 73).**
(XLSX)

**S29 Table. The relative abundance and average composition at the phylum level between groups CCO (n = 134) and CCOS (n = 73).**
(XLSX)

**S30 Table. The abundance and composition at the phylum level of all samples between groups CCO (n = 134) and CCOS (n = 73).**
(XLSX)

**S31 Table. The relative abundance and average composition at the genus level between groups CCO (n = 134) and CCOS (n = 73).**
(XLSX)

**S32 Table. The abundance and composition at the genus level of all samples between groups CCO (n = 134) and CCOS (n = 73).**
(XLSX)

**S33 Table. The different degree of phylum level (p value) between groups CCO (n = 134) and CCOS (n = 73).**
(XLSX)

**S34 Table. The different degree of genus level (p value) between groups CCO (n = 134) and CCOS (n = 73).**
(XLSX)

**S35 Table. The relative abundance and distribution of the key 44 OTUs between groups CCO (n = 134) and CCOS (n = 73).**
(XLSX)

**S36 Table. The cladogram of oropharyngeal microbial structure and their predominant bacteria between groups CCO (n = 134) and CCOS (n = 73).**
(XLSX)

**S37 Table. The corresponding LDA value and p value of microbial community gene function for samples between groups CCO (n = 134) and CCOS (n = 73).**
(XLSX)

**S38 Table. The p value between 27 oropharyngeal OTUs and 6 clinical indicators of CCO (n = 297) and HC (n = 199) obtained by spearman correlation analysis.**
(XLSX)

**S1 Supplementary Methods. Diagnostic, inclusion, and exclusion criteria.**
(DOCX)

## Acknowledgments

We thank all the volunteers who participated in this study, and also thank Hongyan Ren, Jiarui Sun, and Chao Liu (Shanghai Mobio Biomedical Technology Co., Ltd., Shanghai, China) for their help and support.

## Author Contributions

**Conceptualization:** Zujiang Yu, Zhigang Ren.

**Data curation:** Yawen Zou, Liwen Liu, Guizhen Zhang, Benchen Rao.

**Formal analysis:** Ying Sun, Yawen Zou.

**Investigation:** Guangying Cui, Ranran Sun, Yanxia Gao, Xiaorui Liu, Yongjian Zhou, Donghua Zhang, Xueqing Wang.

**Methodology:** Zujiang Yu, Zhigang Ren.

**Project administration:** Guangying Cui, Ranran Sun, Yanxia Gao, Xiaorui Liu, Yongjian Zhou, Xueqing Wang, Yonghong Li, Liwen Liu, Guizhen Zhang, Benchen Rao, Zujiang Yu, Zhigang Ren.

**Resources:** Donghua Zhang, Yonghong Li.

**Writing – original draft:** Ying Sun, Zhigang Ren.

**Writing – review & editing:** Guangying Cui, Ranran Sun, Zujiang Yu, Zhigang Ren.

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
