## [Decision Letter · Decision Letter 0]

30 Oct 2023

Dear RZG Ren,

Thank you very much for submitting your manuscript "Dynamic changes of Bacterial Microbiomes in Oropharyngeal during Infection and Recovery of COVID-19 Omicron Variety" for consideration at PLOS Pathogens. As with all papers reviewed by the journal, your manuscript was reviewed by members of the editorial board and by several independent reviewers. In light of the reviews (below this email), we would like to invite the resubmission of a significantly-revised version that takes into account the reviewers' comments.

The authors should address the reviewers' concerns regarding the comparisons to previous studies and to other variants, potentially by adding further analysis. Also, consider changing the word "variety" to "variant" in the title. The readability of certain tables, e.g. S2 (which might be better suited as a supplementary data set rather than a table) needs to be improved. There are whole pages of table without any contents, so the quality of the supplemental section in general needs to be improved to publication quality. Claims of "first time" should be removed from the discussion. Much of the speculation on mechanisms in the discussion is unsupported and should be toned down in the absence of any supporting data. Further evidence towards the influence of the short chain fatty acids derived from these bacteria on SARS-CoV-2 infection severity would improve the manuscript.

We cannot make any decision about publication until we have seen the revised manuscript and your response to the reviewers' comments. Your revised manuscript is also likely to be sent to reviewers for further evaluation.

Sincerely,

Ashley L. St. John

Academic Editor

PLOS Pathogens

Ana Fernandez-Sesma

Section Editor

PLOS Pathogens

Kasturi Haldar

Editor-in-Chief

PLOS Pathogens

orcid.org/0000-0001-5065-158X

Michael Malim

Editor-in-Chief

PLOS Pathogens

orcid.org/0000-0002-7699-2064

The authors should address the reviewers' concerns regarding the comparisons to previous studies and to other variants, potentially by adding further analysis. Also, consider changing the word "variety" to "variant" in the title. The readability of certain tables, e.g. S2 (which might be better suited as a supplementary data set rather than a table) needs to be improved. There are whole pages of table without any contents, so the quality of the supplemental section in general needs to be improved to publication quality. Claims of "first time" should be removed from the discussion. Much of the speculation on mechanisms in the discussion is unsupported and should be toned down in the absence of any supporting data. Further evidence towards the influence of the short chain fatty acids derived from these bacteria on SARS-CoV-2 infection severity would improve the manuscript.

Reviewer's Responses to Questions

**Part I - Summary**

Reviewer #1: In this study the authors characterize oropharyngeal microbiomes of individuals infected by the Omicron variant to show that their microbiome vary from that of the original strain. The authors state that this is important because the oropharyngeal microbiome plays an important role in the severity of the disease. They then go on to construct a panel of a non-invasive microbial markers which could potentially be used for diagnostic purposes.

The sample size of this study is clearly a strength however the novelty and significance are lowered given that another study by Zhang et al., also show (albeit with a smaller sample size) that the oropharyngeal microbiome diversity is reduced and less diverse in SARS-CoV2 patients. The novelty of the study, as stated by the authors, is that the Omicron variant had different microbiomes from the original strain. They also include a cohort of patients that have recovered from the Omicron variant.

In my opinion the study is executed well (the authors also previously publish this study design for the original strain of SARS-CoV2, Gao et al., 2021). However, I think the strength of the study could be improved by adding further mechanistic study to account for the differences in oropharyngeal microbiomes of individuals infected with Omicron.

Reviewer #2: Summary:

The article delves into the role of the oropharyngeal microbiome in patients infected with the COVID-19 Omicron variant. The researchers identify significant microbial imbalances in the oropharynx of infected individuals compared to healthy controls, developing a microbial-based diagnostic model, and highlighting unique microbiome characteristics between Omicron and the original strain.

Key findings indicate an increase in conditional pathogens like Prevotella and Veillonella in the infected group, while acidogenic bacteria such as Fusobacterium and Alloprevotella, vital for metabolic functions, decreased.

The study also developed a diagnostic model based on oropharyngeal microbial markers, demonstrating a high accuracy rate in distinguishing Omicron infections. Upon examining recovered Omicron patients, the study identified microbial changes that could influence post-infection conditions. Notably, lactic acid's role in immune response and Atopobium's potential involvement in inflammation were highlighted. Comparisons between the Omicron strain and the original strain showed unique oropharyngeal microbiome characteristics.

Finally, the study connected clinical indexes with the oropharyngeal microbiome, like the negative correlation between neutrophils and the bacteria Prevotella.

The study contributes new knowledge about how oropharyngeal microbiomes change during infection and recovery, which could be valuable for understanding disease progression and recovery.

Comments:

Scope and Relevance: The investigation into the oropharyngeal microbiome's role in the context of COVID-19 infections, particularly with the Omicron variant, is both timely and relevant. As the pandemic evolves and new strains emerge, understanding these nuances becomes crucial for diagnostics and therapeutic interventions.

Sample Size and Diversity: The study boasts a relatively large sample size. However, the distribution among categories appears skewed. For instance, there's a discrepancy between confirmed cases of Omicron (297) versus the original strain (73), which might influence the comparative outcomes.

Use of MiSeq Sequencing: MiSeq sequencing provides detailed insights into microbial communities and is a validated method for the study of microbiomes.

Key Findings: The observed microbial imbalance—increased conditional pathogens and decreased acidogenic bacteria—provides a noteworthy insight into potential secondary infections or complications during Omicron infections. Yet, the study could benefit from further elaborating on the direct consequences and underlying mechanisms of these microbial shifts.

Diagnostic Model: The development of a diagnostic model based on oropharyngeal microbial markers is innovative. However, its real-world application would necessitate validation across varied populations and more extensive sample sizes to ensure generalizability.

Limitations Acknowledgment: While the study acknowledges its limitations—like the inability to stratify disease severity and the lack of animal model validation—it could have benefited from addressing potential confounding factors, like the effect of concurrent treatments on the microbiome, dietary differences, or geographical variations.

Causation vs. Correlation: While the study found associations between oropharyngeal microbiomes and COVID-19 Omicron variant infection, it does not necessarily mean the changes in microbiomes caused or were caused by the infection. Further research is required to establish causality under the light of confounding factors. Like whether the patients were taking general over the counter medications to mitigate fever and other symptoms etc. before confirmation of covid infection (omicron or not). This may affect the microbiome quite significantly.

Conditional Pathogens: The term "conditional pathogens" is vague. It would be beneficial for readers if the specific pathogens were listed, along with how they might be related to COVID-19 or other infections.

Potential Confounders: There might be other factors influencing the oropharyngeal microbiome (e.g., diet, antibiotic usage, OTC drugs, other infections etc.). It's essential to consider and account for these potential confounders in the analyses.

The phrase "significant difference" is used multiple times, especially in the context of NMDS and PCoA analyses in Section 2.2. It would be helpful to elaborate on what kind of differences were observed. Were certain bacterial taxa more prevalent in one group than the other? Were there shifts in the overall microbial community composition?

Discussion mentions “NET formation”, which presumably refers to Neutrophil Extracellular Traps. Given the importance of this mechanism in the context of infections, it might be beneficial to explain it briefly.

It is unclear how it may lead to potential clinical implications and applications of the findings. How can these results be used to improve patient care or guide future research? Worth discussing.

The term "packet" in "R packet" (Section 4.5) is typically referred to as "package". Consider revising to "R package.

The section 4.5 about the cutoff value for the LDA score might benefit from a rationale: Why were these particular cutoffs chosen?

The term Oropharynx should be used in the title and elsewhere instead of oropharyngeal, wherever relevant. The manuscript requires a comprehensive review for linguistic accuracy and clarity in its expression.

**Part II – Major Issues: Key Experiments Required for Acceptance**

Reviewer #1: 1. Have the authors read Ma, S., Zhang, F., Zhou, F. et al. Metagenomic analysis reveals oropharyngeal microbiota alterations in patients with COVID-19. Sig Transduct Target Ther 6, 191 (2021). https://doi.org/10.1038/s41392-021-00614-3. This is not sited in the current version of the manuscript and should be given the similarities of this study with the authors current manuscript. The authors should also add in a discussion comparing the findings of both papers.

2. Can the authors sample patients with another variant of less severity to strengthen the notion that these are bonafide markers of the Omicron variant?

3.I find this manuscript is very similar to their previous work on the original strain Gao et al., 2021 and I feel that this manuscript would be strengthened by further research into the molecular mechanism and/or possible pathways and targets of the microbiome which can be related to the Omicon clinical severity or outcome.

Reviewer #2: Analysis of the data under the light of confounding factors and incorporation of more healthy controls for better comparisons

Association of the findings with disease severity or response to treatment. Whether the patients were vaccinated and their response/time of vaccination should be considered.

More clarity is needed on the 'diagnostic model of oropharyngeal microbial markers' perhaps comparing with the disease outcomes and response to drugs/vaccines. Clinical potential seems to be there but not clear.

**Part III – Minor Issues: Editorial and Data Presentation Modifications**

Reviewer #1: 1. bacterial strains should be italicized.

2. Figure 3 legends should include a separate reference to each graph within figure 3

Reviewer #2: The legends accompanying the figures could benefit from further elaboration, including the addition of specific experimental details and the statistical tests employed. Additionally, the clarity and presentation of the figures themselves should be enhanced, as their current quality appears to be compromised.

PLOS authors have the option to publish the peer review history of their article (what does this mean?). If published, this will include your full peer review and any attached files.

Reviewer #1: No

Reviewer #2: No
---

## [Decision Letter · Decision Letter 1]

19 Feb 2024

Dear RZG Ren,

Thank you very much for submitting your manuscript "Dynamic changes of Bacterial Microbiomes in Oropharynx during Infection and Recovery of COVID-19 Omicron Variant" for consideration at PLOS Pathogens. As with all papers reviewed by the journal, your manuscript was reviewed by members of the editorial board and by several independent reviewers. The reviewers appreciated the attention to an important topic. Based on the reviews, we are likely to accept this manuscript for publication, providing that you modify the manuscript according to the review recommendations.

The reviewers were satisfied with the major changes that were made to the figures and text with revision and considered the manuscript to be improved and to be of broad interest. Minor clarifications were requested by Reviewer 1 which should be considered by the authors to improve the clarify.

Sincerely,

Ashley L. St. John

Section Editor

PLOS Pathogens

Ana Fernandez-Sesma

Section Editor

PLOS Pathogens

Michael Malim

Editor-in-Chief

PLOS Pathogens

orcid.org/0000-0002-7699-2064

The reviewers were satisfied with the major changes that were made to the figures and text with revision and considered the manuscript to be improved and to be of broad interest. Minor clarifications were requested by Reviewer 1 which should be considered by the authors to improve the clarify.

Reviewer Comments (if any, and for reference):

Reviewer's Responses to Questions

**Part I - Summary**

Reviewer #1: the authors have done a good job responding to the reviewers comments

Reviewer #2: Authors have done a good job at comparative analysis in the revised manuscript and have answered all the questions raised during the review.

**Part II – Major Issues: Key Experiments Required for Acceptance**

Reviewer #1: (No Response)

Reviewer #2: Authors have answered all my questions with satisfactory explanation and evidences through analysis.

**Part III – Minor Issues: Editorial and Data Presentation Modifications**

Reviewer #1: line 46 restructure sentence to something like "Oropharyngeal microbiomes play a significant role in the susceptibility and severity of COVID-19, yet the role these microbiomes play for the development of COVID-19 omicron variant have not been reported.

line 53 define the best

line 54 define OTU

Reviewer #2: (No Response)

PLOS authors have the option to publish the peer review history of their article (what does this mean?). If published, this will include your full peer review and any attached files.

Reviewer #1: **Yes: **kim samirah robinson

Reviewer #2: **Yes: **Archita Mishra

Figure Files:

Data Requirements:

Reproducibility:

References:

---

## [Editor Report · Decision Letter 2]

26 Feb 2024

Dear RZG Ren,

We are pleased to inform you that your manuscript 'Dynamic changes of Bacterial Microbiomes in Oropharynx during Infection and Recovery of COVID-19 Omicron Variant' has been provisionally accepted for publication in PLOS Pathogens.

Best regards,

Ashley L. St. John

Section Editor

PLOS Pathogens

Ana Fernandez-Sesma

Section Editor

PLOS Pathogens

Michael Malim

Editor-in-Chief

PLOS Pathogens

orcid.org/0000-0002-7699-2064
---

## [Editor Report · Acceptance letter]

15 Mar 2024

Dear RZG Ren,

We are delighted to inform you that your manuscript, "Dynamic changes of Bacterial Microbiomes in Oropharynx during Infection and Recovery of COVID-19 Omicron Variant," has been formally accepted for publication in PLOS Pathogens.

Best regards,

Michael Malim

Editor-in-Chief

PLOS Pathogens

orcid.org/0000-0002-7699-2064